# Synthetic CB1 Cannabinoids Promote Tunneling Nanotube Communication, Cellular Migration, and Epithelial–Mesenchymal Transition in Pancreatic PANC-1 and Colorectal SW-620 Cancer Cell Lines

**DOI:** 10.3390/cells14020071

**Published:** 2025-01-07

**Authors:** David A. Bunsick, Leili Baghaie, Yunfan Li, Abdulrahman M. Yaish, Emilyn B. Aucoin, Elizabeth Skapinker, Rashelle Aldbai, Myron R. Szewczuk

**Affiliations:** 1Department of Biomedical & Molecular Sciences, Queen’s University, Kingston, ON K7L 3N6, Canada; david.bunsick@umassmed.edu (D.A.B.); 16lbn1@queensu.ca (L.B.); 19ra57@queensu.ca (R.A.); 2Faculty of Arts and Science, Queen’s University, Kingston, ON K7L 3N9, Canada; 18yl210@queensu.ca (Y.L.); 21ess18@queensu.ca (E.S.); 3Faculty of Health Sciences, Queen’s University, Kingston, ON K7L 3N9, Canada; a.yaish@queensu.ca; 4Faculty of Science, Biology (Biomedical Science), York University, Toronto, ON M3J 1P3, Canada; emilynaucoin@gmail.com

**Keywords:** CB1 cannabinoids, tunneling nanotube, migration, EMT, pancreatic PANC-1, colorectal SW-620, inhibitor OP

## Abstract

Our recent findings unveil a potential game-changer in cancer therapy. We have discovered that synthetic CB1 cannabinoids, specifically AM-404, arvanil, and olvanil, can surreptitiously orchestrate tunneling nanotubes, enhance cell viability, and promote scratch wound migration and epithelial–mesenchymal transition (EMT) in pancreatic PANC-1 and colorectal SW-620 cancer cells.

## 1. Introduction

Synthetic cannabinoids, such as AM-404, arvanil, and olvanil, interact uniquely with the endocannabinoid system (ECS), particularly targeting CB1 receptors. CB1, a class A G protein-coupled receptor (GPCR), is overly expressed in the central nervous system (CNS) and plays a crucial role in various physiological processes [1,2]. The CB1 GPCRs are also expressed in the periphery but at much lower levels than in the central nervous system (CNS) [3]. Recent studies indicate that CB1 GPCRs are expressed in pancreatic [4] and colorectal [5] cancer cells. Bunsick et al. [6] reported that CB1 agonists can influence both pancreatic and colorectal cancer cellular behavior by modulating GPCR signaling pathways via a functional selectivity mechanism(s), which was found to be essential in regulating cell viability, migration, and epithelial–mesenchymal transition (EMT). Cherkasova et al. [7] reported an excellent review of the cannabinoid system in regulating normal and inflamed intestines, the colorectal cancer properties of CB1 and CB2 receptor cannabinoids, and their role in colorectal cancer pathogenesis, prevention, and treatment. In primary cell cultures, CB1 and CB2 receptors have been reported to be expressed at a higher level in pancreatic β cells compared with non-β cells [7]. Interestingly, in addition to CB1 and CB2, there are several other receptors, such as GPR119, GPR55 [8], peroxisome proliferating activated receptor *a* (PPAR*a*), and PPAR*g* [9], which cannabinoids may respond to [10]. Bosier et al. [11] reported on the diversity of cannabinoid-mediated signaling and how the diverse mechanisms allow for specificity in the associated functional responses triggered by endogenous or exogenous ligands.

CB1 receptors can also form heterodimers to increase their potential to generate biased downstream signaling effects. For example, CB1 can heterodimerize with various other GPCRs, including the D2 dopamine receptor [12], µ-opioid receptor [13], A2A adenosine receptor [14], and β2 adrenergic receptors (β2AR) [15]. Overall, heterodimerization significantly increases the complexity of CB1 functional selectivity as it depends not only on the identities of the GPCRs present in the complex but also on the active states of each of the receptors [15].

Research on cannabinoids has also evolved significantly in revealing their complex role in cancer progression and metastasis. For example, Yingzhi et al. [16] reported, in terms of migration and invasion, that the cannabinoid delta-9-tetrahydrocannabinol (Δ9-THC) can inhibit these processes by reducing matrix metalloproteinase (MMP) activity and downregulating EMT markers, such as vimentin and N-cadherin. This process is the opposite of our previously reported finding that synthetic cannabinoids have been shown to increase EMT markers in pancreatic cancer cells [6]. The potential reason(s) is that Δ9-THC is the main psychotropic component of cannabis and is an agonist of both CB1 and CB2 receptors [17]. In addition, cannabinoids can induce cell cycle arrest at the G1 phase in cancer cells by upregulating p21 and downregulating cyclin D1, thereby inhibiting cell proliferation and viability [18]. Ellert-Miklaszewska et al. [19] also reported that synthetic cannabinoids can promote apoptosis through the mitochondrial pathway by increasing Bax/Bcl-2 ratios and activating caspase-3.

To address these cannabinoid reaction complexities, Bunsick et al. [6] recently introduced the concept of biased agonism or functional selectivity in CB1 GPCRs. Synthetic CB1 cannabinoids have been shown to induce NF-κB-dependent secreted alkaline phosphatase (SEAP) activity, leading to the expression of EMT markers, such as E-cadherin and vimentin, in colorectal SW-629 cancer cells [6]. NF-κB is a key inflammatory transcription factor involved in immune responses and tumor progression [20]. Its activation leads to the expression of genes that promote inflammation, survival, proliferation, and metastasis [20]. Activation of NF-κB has been shown to activate epigenetic factors that might transmogrify cancer metabolism and epigenetic reprogramming to a metastatic phenotype [21]. CB1-mediated changes in histone acetylation can enhance the expression of NF-κB target genes, thereby promoting tumor progression [21]. Specifically, CB1 activation can increase the activity of histone acetyltransferases (HATs), with acetylated histones at the promoters of NF-κB target genes [21,22]. This acetylation reduces the positive charge on histones, decreasing their interaction with the negatively charged DNA and resulting in a relaxed chromatin structure [23]. This relaxed structure allows transcription factors, including NF-κB, to more easily access the DNA and initiate transcription [24]. As a result, genes involved in cell survival, proliferation, and inflammation are upregulated, contributing to tumor progression [24].

The activation of NF-κB by CB1 agonists involves the heterodimerization of CB1 GPCRs with neuromedin B GPCRs (NMBRs) in complex with Neu-1 and MMP-9 [6]. This CB1 heterodimerization with NMBRs activates Neu-1, which can alter the glycosylation state of receptors, thereby modulating downstream signaling pathways that contribute to cancer progression [6]. This unique CB1 heterodimerization process affects the glycosylation state of the receptor, modulating downstream signaling and activating glycosylated receptor tyrosine kinases (RTKs) and Toll-like receptors (TLRs) [6,25]. These findings underscore the complexity and therapeutic potential of CB1 receptor signaling, necessitating further research into their mechanistic actions.

In the present study, we investigated the effects of synthetic cannabinoids on orchestrating tunneling nanotubes (TNTs) to enhance metabolic cellular driver viability affecting scratch wound migration and epithelial–mesenchymal transition (EMT) in pancreatic PANC-1 and colorectal SW-620 cancer cells. TNTs are intercellular conduits used by normal, immune, and cancer cells to facilitate the transfer of calcium waves, mitochondria, lysosomes, and proteins. TNTs, in turn, have been reported to contribute to survival, metastasis, and chemoresistance in cancer cells [26]. Here, we hypothesize that CB1 receptor agonists promote TNTs to regulate metabolic driver cellular viability via the Neu-1-MMP9 signaling axis, thereby inducing cancer metastasis. By examining these factors, we seek to understand cancer cell communication behaviors and the potential risk of cannabinoid use in cancer therapy.

## 2. Materials and Methods

### 2.1. Cell Lines

Three cell lines were used in this study: PANC-1 (ATCC^®^ CRL-1469™), SW-620 (ATCC^®^ CCL-227™), and RAW-Blue macrophages (InvivoGen, San Diego, CA, USA) cells. RAW-Blue™ cells (Mouse Macrophage Reporter Cell Line, InvivoGen) derived from RAW264.7 macrophages were grown in a culture medium containing the selectable marker Zeocin [27]. They stably express a secreted embryonic alkaline phosphatase (SEAP) gene inducible by AP-1 and NF-κB transcription factors. RAW-Blue™ cells, upon stimulation, activate NF-κB and AP-1, leading to SEAP secretion, which is detectable and measurable using QUANTI-Blue™ (InvivoGen) SEAP in the medium. RAW-Blue™ cells are made to be resistant to Zeocin™ and G418 antibiotics. The cells were grown in conditioned media 1× Dulbecco’s modified eagle medium (DMEM, Gibco, Rockville, MD, USA), with 10% fetal bovine serum (FBS) (HyClone, Logan, UT, USA) and 5 μg/mL plasmocin (InvivoGen, San Diego, CA, USA) and maintained at 5% CO_2_ and 37 °C.

### 2.2. Reagents

The CB1 agonists N-arachidonoyl aminophenol (AM-404) (EC50: 2.57 μg/mL) [28], arvanil (EC50: 1.28 μg/mL) [29], and olvanil (EC50: 4.127 μg/mL) [29] were purchased from Alomone Labs, Jerusalem Bio Park (JBP) (Jerusalem, Israel) and utilized in a dose-dependent manner. The EC50 value was used to create the dose–response curve. The saturation value was a 3:1 concentration of the EC50 value for each agonist. The subsequent doses were 1:10 and 1:30 dilutions of the EC50 value.

The neuromedin B receptor (NMBR) inhibitor BIM-23127 was used at 20.9 nM and purchased from Tocris Bioscience, Moorend Farm Avenue, Bristol, BS11 0QL, UK. Oseltamivir phosphate (OP) (>99% pure OP, batch No. MBAS20014A, Solara Active Pharma Sciences Ltd., New Mangalore-575011, Karnataka, India), a broad-range neuraminidase sialidase inhibitor used at predetermined effective dosages, was used at 300 μg/mL. An MMP-9 inhibitor (MMP-9-i), a potent reversibly selective inhibitor of MMP-9, was used at 5 nM and purchased from GLPBIO Technology, 10292 Central Ave #205, Montclair, CA 91763, USA.

### 2.3. CB1 Agonist Treatment Protocol (Time and Dosage)

The agonists AM-404, arvanil, and olvanil were chosen based on their interactions with the CB1 receptor and binding affinity. AM-404 is an indirect agonist of the CB1 receptor by binding to the fatty acid amide hydrolase (FAAH) enzyme, preventing anandamide breakdown [30,31]. AM-404 has an EC50 value of ~6.5 µM (2.57 µg/mL) [32,33,34,35]. Arvanil is a potent direct agonist of the CB1 receptor and prevents anandamide breakdown. The agonist has an EC50 value of ~2.9 µM (1.27 µg/mL) [30,35,36,37]. Olvanil is also a direct agonist of the CB1 receptor. Olvanil is a weaker agonist with an EC50 value of ~10.1 µM (4.21 µg/mL) [35,38,39]. These values were used to create a dose–response curve where the EC50 value is a 1:3 dilution of the saturation dose. Dilutions of 1:10 and 1:30 were made from the saturation dose. The cells were treated with the synthetic cannabinoid for 24 h, as previously reported by others [40,41,42].

### 2.4. Antibodies

Primary mouse, monoclonal IgG antibodies for E-cadherin and N-cadherin were purchased from Santa Cruz Technologies, Inc. (Dallas, TX 75220, USA) and used at a 1:10 dilution from 200 μg stock for immunofluorescence staining. The secondary goat anti-mouse Alexa Fluor 488 antibodies (Santa Cruz Technologies) were used at a concentration of 1:1000 for the immunofluorescence predetermined standardized protocol.

### 2.5. Immunofluorescence Staining

PANC-1 cells were cultured and individually grown in a sterile 24-well plate on 12 mm circular glass coverslips containing the conditioned medium for 24 h. Upon reaching approximately 70% confluence on the glass slide, the cells were media-starved and incubated with predetermined and indicated concentrations of CB1 agonists (AM 404, olvanil, and arvanil) in the designated wells for 24 h. The control wells were incubated with media without FBS. The cells were then fixed at 4% PFA (300 µL) and incubated in the cold for 24 h. To facilitate the binding of antibodies, the cells were permeabilized for 5 min with 0.2% TritonX 100 (300 µL) and then blocked with 4% BSA/Tween20/TBS (300 µL) for 24 h in the cold to prevent non-specific binding. Primary antibodies were added at a 1:10 dilution in a 4% BSA/Tween20/TBS solution and incubated in the fridge for 24 h. Subsequently, the secondary antibodies were added at 1:, Ver1000 dilution in PBS and incubated in the fridge for 24 h. Cells treated with secondary antibodies were employed as a negative control. Stained cells on coverslips were mounted with 3 μL DAPI mounting media. Slide images were observed using Zeiss Imager M2 epi-fluorescent microscopy (20× magnification, Carl Zeiss Canada Ltd., M3B 2S6 Toronto, ON, Canada), capturing images under the Alexa 488 channel. The fluorescence of each marker was analyzed using Corel Photo-Paint, Version 23.1.0.389, where an average of eight points on the image was made and subtracted by background fluorescence, and the difference was multiplied by the pixel density.

### 2.6. Tunneling Nanotubes (TNTs)

PANC-1 cells were cultured and individually grown in a sterile 24-well plate on 12 mm circular glass coverslips containing the conditioned medium for 24 h. The cells were media-starved and incubated with predetermined and indicated concentrations of the CB1 agonists (AM-404, arvanil, and olvanil) in the designated wells for 24 h after reaching 70% confluency on the glass slide. For the TNT assay using OP, cells were treated with 300 µg/mL OP for 15 min before synthetic cannabinoid treatment. The control wells were incubated with media without FBS. The cells were washed once with PBS, treated with Invitrogen CellMask Plasma Membrane Stain (C10045, Invitrogen, Thermo Fisher Scientific Inc., Waltham, MA, 02451, USA) and fixed at 4% PFA (300 µL) before being incubated in 4 °C for 24 h. Following this, the cells were washed 3 times with PBS-T and mounted on microscope slides using 3 μL of VECTASHIELD DAPI fluorescent mounting medium (VECTH1500, M.J.S. BioLynx Inc. 300 Laurier Blvd, K6V 5W1, Brockville, ON, Canada). The slide images were observed using Zeiss M2 epi-fluorescent microscopy (20× magnification, Carl Zeiss Canada Ltd., M3B 2S6 Toronto, ON, Canada), capturing images under the Rhodamine (554 nm) channel. The images were enhanced, and the cell projections were differentiated using ImageJ, Version 23.1.0.389. The projections were quantified using Corel Photo-Paint, Version 23.1.0.389. For statistical analysis, we used GraphPad Prism 10. Comparisons between groups from two independent experiments were conducted using a one-way analysis of variance (ANOVA) at 95% confidence, followed by Fisher’s uncorrected LSD post hoc test with 95% confidence. Asterisks denote statistical significance.

### 2.7. AlamarBlue Cell Viability Assay

PANC-1 and SW620 cells were plated with ~20,000 cells per well in a flat-bottom 96-well plate containing culture medium for 24 h at 37 °C and 5% CO_2_. Following incubation, the cells were treated with 90 μL of each CB1 agonist in a medium without FBS for 24 h alone or in combination with the three inhibitors for 15 min before ligand stimulation. Next, 10 μL of AlamarBlue reagent was added to each well and incubated for 4 h. The absorbance was recorded using a spectrophotometer (Spectra Max 250, Molecular Devices, Sunnyvale, CA 94089, USA) at 570 nm. Each experiment was performed in triplicate. Cell viability was determined based on the fluorescence from the blank media control and the untreated cellular control. The following formula was used to determine cell viability as a percent of the control for each drug concentration: [(Absorbance of cells in a drug concentration) − (Media absorbance)]/[(Absorbance of cells alone) − (Media absorbance)] × 100.

### 2.8. NF-kB Dependent Secreted Embryonic Alkaline Phosphatase (SEAP) Assay

Briefly, a cell suspension of 1 × 10^6^ cells/mL in the fresh growth medium was prepared, and 100 μL of a RawBlue suspension of cells (~100,000 cells) was added to each well of a Falcon flat-bottom 96-well plate (Becton Dickinson, Mississauga, ON, Canada) [27]. Following varying incubation times, AM-404, olvanil, and arvanil CB1 agonists were added to each well in a dose-dependent manner, either alone or in combination with the specific MMP-9 inhibitor (MMP-9i). Oseltamivir phosphate (OP) and BIM-23127 (BIM-23) were added to each well 15 min before stimulation with ligands. The experimental plates were incubated in 5% CO_2_ at 37 °C for 18–24 h, followed by QUANTI-Blue™ (InvivoGen) reagent solution per the manufacturer’s instructions. Briefly, 160 μL of resuspended QUANTI-Blue solution was added to each well of a 96-well flat-bottom plate, adding 40 μL supernatant from the treated RAW-blue cells. Following the plate incubation for 60 min at 37 °C, the SEAP levels were measured using a spectrophotometer (Spectra Max 250, Molecular Devices, Sunnyvale, CA 94089, USA) at 620–655 nm. Each experiment was performed in triplicate.

### 2.9. Scratch Wound Assay

PANC-1 and SW-620 cells were grown in a 50 mm glass bottom culture dish (MatTek, 200 Homer Avenue—Ashland, MA, 01721, USA) in a culture medium and allowed to adhere to 90% confluence on the glass slide in an incubator at 37 °C and 5% CO_2_. A sterile pipette tip created a wound, and non-adherent cells were removed using a medium wash. The control cells were then supplemented with a medium containing 5% FBS. At the same time, the CB1 agonist-treated cells were supplemented with AM-404, arvanil, or olvanil in a medium with 5% FBS. Imaging was taken with a Nikon eclipse Ti2 microscope (4× magnification, Nikon Instruments Inc., Melville, NY, USA) hourly for the first 6 h, then at 12 after the creation of the wound. The wound width was measured at 6–8 points per image using NIS-Elements AR software, Version 5.21.00 to create a simple linear regression using GraphPad Prism 10, demonstrating the rate of wound closure represented by μm/h.

### 2.10. Statistics

For statistical analysis, we used GraphPad Prism 10. Comparisons between groups from two independent experiments were conducted using a one-way analysis of variance (ANOVA) at 95% confidence, followed by Fisher’s uncorrected LSD post hoc test with 95% confidence. Asterisks denote statistical significance.

## 3. Results

### 3.1. Synthetic CB1 Cannabinoids AM-404, Arvanil, and Olvanil Significantly Alter Cancer Cell Phenotype and Induce Tunneling Nanotubes (TNTs) in PANC-1 Cells

To assess the impact of the synthetic CB1 cannabinoids on cancer cells, we investigated the ability of AM-404, arvanil, and olvanil at their EC50 concentration to form PANC-1 tunneling nanotubes (TNTs) after 24 h in culture. We performed an assay using a CellMask membrane stain with an excitation of 554 nm. The results indicate that treating PANC-1 cells with AM-404, arvanil, and olvanil significantly increased the number of cellular projections compared to the untreated control (Figure 1A, TNT images). The increase in the number of cellular projections can also be visualized in Figure 1B CB1 treatment and Figure 1D–F (TNT images) following OP treatment, where small, thin projections exist between neighboring cells and across longer distances. Additionally, the cell phenotype following synthetic cannabinoid treatment differs from the untreated control. The untreated control appears rounder, with fewer projections that do not span longer distances.

On the other hand, the synthetic cannabinoid-treated cells have a more jagged appearance around the edges and are longer compared to the untreated control. When cells are treated with oseltamivir phosphate (OP) at 300 µg/mL, there is a significant reduction in cellular projections compared to the treatment with the synthetic cannabinoid alone (Figure 1D–F). Additionally, the phenotype of the cells became rounder and smaller, similar to that of the untreated control. However, the untreated control cells were larger compared to the cells treated with OP (Figure 1C). The data depicted in Figure 1 support their communications for invasion induced with AM-404, arvanil, and olvanil. Notably, there are changes in the shape of the treated cell compared to the untreated control. The control group is rounder, while the treated cells are more jagged in shape. When cancer cells metastasize, they morph, becoming missile-like in shape so that they can penetrate other tissues throughout the body.

### 3.2. Synthetic CB1 Cannabinoids AM-404, Arvanil, and Olvanil Significantly Increase the Metabolic Activity of Pancreatic PANC-1 and Colorectal SW-620 Cells

Noteworthy, the data depicted in Figure 1 demonstrate that the synthetic CB1 cannabinoids AM-404, arvanil, and olvanil significantly impacted the pancreatic PANC-1 cancer cell phenotype to initiate their communications for invasion. Interestingly, Bergers and Fendt [43] reported that metastasizing cancer cells can surreptitiously adapt their metabolic activity during their metastatic invasion. Also, cancer metabolic activity is significantly related to chemoresistance [44,45,46]. By initiating their communications for invasion, cancer cells can reprogram their cellular activities to initiate their proliferation and migration and uniquely counteract metabolic stress during their progression [46]. During this reprogramming process, cancer cells’ metabolism and other cellular activities are integrated and mutually regulated by tunneling nanotube communications to alter their specific metabolic functional drivers of tumor growth and progression [46].

To this end, we investigated the metabolic activity of PANC-1 and SW-620 cells following treatment with the synthetic CB1 cannabinoids AM-404, arvanil, and olvanil at the predetermined EC50 values from Figure 1. To test this metabolic activity, we used the AlamarBlue cell viability assay. The AlamarBlue reagent is an indigo-colored, non-toxic resazurin-based solution that indicates cell health by using the reducing power of living cells to measure viability quantitatively. Upon entering living cells, resazurin is reduced to resorufin. This red compound is highly fluorescent and provides accurate time–course measurements of the metabolic activity of healthy cells with high sensitivity and linearity. It involves no cell lysis [47]. The data depicted in Figure 2 reveal that the synthetic cannabinoids significantly increased PANC-1 cell viability and metabolic activity compared to the untreated and positive (media containing FBS) controls. As seen in Figure 2A, AM-404 consistently increased the cell viability the most compared to the other agonists. At the maximum concentration, arvanil has increased cell viability compared to olvanil; however, the difference between the two was insignificant (Figure 2B,C). For the SW-620 cell line, all three synthetic cannabinoids significantly upregulated cell viability compared to the untreated control (Figure 3A–C).

Interestingly, AM-404 had the weakest impact on cell viability compared to arvanil and olvanil. Additionally, both AM-404 and olvanil exhibited a dose-dependent response relationship but increased viability with lower concentrations, similar to the PANC-1 cells. However, arvanil did not display a dose–response relationship.

Recently, Bunsick et al. [6] reported that CB1 cannabinoid GPCRs form a complex with a novel biased GPCR signaling paradigm involving the crosstalk between neuraminidase-1 (Neu-1) and matrix metalloproteinase-9 (MMP-9) in the activation of glycosylated receptor tyrosine kinases (RTKs) and Toll-like receptors (TLRs) through the modification of the receptor glycosylation state for downstream signaling. To determine if inhibiting this novel CB1 GPCR signaling platform affects cell viability and metabolic activity, the cancer cells were treated with the NMBR inhibitor BIM-23127 (BIM-23), MMP-9 inhibitor I (MMP9-i), a potent reversibly selective inhibitor of MMP-9, and oseltamivir phosphate (OP), a broad range neuraminidase sialidase inhibitor, for 15 min, followed by the synthetic cannabinoids. We treated PANC-1 and SW-620 cells with either AM-404, arvanil, or olvanil alone or in combination with the inhibitors (BIM-23, MMP-9i, or OP) at the indicated dosage for 24 h. The addition of synthetic CB1 cannabinoids significantly increased cell viability compared with the untreated control for PANC-1 cells (Figure 2A–C) and SW-620 cells (Figure 3A–C). When they were combined with the inhibitors, the cell viability decreased significantly with all three synthetic cannabinoids and the positive control (media containing 10% FBS) (Figure 2D–F, H and Figure 3D–F,H). Cell viability of the PANC-1 cells was not affected by the CB1 cannabinoids with the untreated controls (Figure 2G), while the SW-620 cells were slightly affected. Additionally, cell viability for both cells remained at 80–95% with the inhibitors alone, similar to the untreated control, indicating that using these inhibitors did not result in non-specific adverse cell death (Figure 2G), but the SW-620 cells were slightly affected (Figure 3G).

### 3.3. Synthetic CB1 Cannabinoids AM-404, Arvanil, and Olvanil Significantly Increase the Metastatic EMT Expression of N-Cadherin in Pancreatic PANC-1 Cancer Cells

Bunsick et al. [6] reported on the EMT expression markers of SW-620 cells. They found that SW-620 cells showed no statistically significant reduction in E-cadherin following treatment with the CB1 cannabinoids AM-404, arvanil, and olvanil when compared to the control group, suggesting that SW-620 cells had already undergone the EMT process for invasiveness. The SW620 cell line was derived from a colon cancer lymph node metastasis (Duke’s type C) and had a rounded or spindle-shaped fibroblast-like morphology [48]. To investigate the effect of the synthetic CB1 cannabinoids on E- and N-cadherin expressions in pancreatic PANC-1 cancer cells, we first investigated the presence of E-cadherin following treatment, similarly to the SW-620 cells. PANC-1 cancer cells are derived from the primary metastatic tumor of a patient [49]. PANC-1 is an adenocarcinoma in the head of the pancreas, which invades the duodenal wall. Metastases in one peripancreatic lymph node are deemed evidence during a pancreaticoduodenectomy. Following a 24 h treatment with the synthetic AM-404, arvanil, and olvanil cannabinoids, the data depicted in Figure 4A, B show that each of the CB1 cannabinoids significantly reduced E-cadherin expression compared to untreated control. Here, further investigation is required to elucidate the characteristic properties of metastatic cancer cells influenced by the activation of cannabinoids via their functional selectivity [6].

On the other hand, N-cadherin, like E-cadherin, is a classical cadherin involved in cell–cell adhesion, cell structure maintenance, and cell proliferation [50]. However, an increase in N-cadherin is considered a hallmark of metastatic EMT phenotype, as it promotes tumor cell migration and proliferation by facilitating tumor–host contact with N-cadherin-expressing cells [51,52]. Given that treatment with synthetic CB1 cannabinoids decreased E-cadherin in PANC-1 cells (Figure 4A,B) and increased vimentin in SW-620 cells, as previously reported [6], the data in Figure 4C,D show that CB1 cannabinoids significantly increased N-cadherin compared to the untreated control, suggesting CB1 cannabinoids induce the process of metastatic phenotypes.

It is noteworthy that CB1 agonist-induced internalization of G-protein-coupled receptors is a regulatory mechanism for CB1 receptor abundance and availability at the plasma membrane [53,54]. CB1 receptor trafficking is dynamically regulated by cannabimimetic drugs whereby the recovery of CB1 to the cell surface after 20 min but not 90 min after agonist treatment was reported to be independent of new CB1 receptor synthesis [54]. As depicted in Figure 4, the pattern of expression and the localization of the E- and N-cadherins following CB1 agonist treatment for 24 h is elucidated by fixing and permeabilizing the cells to detect the proteins expressed at the cell plasma membrane (cell junction functional) and in the cytoplasm (cell junction degraded and not functional). Glogauer and Blay [55] reviewed the dynamic diversity in cancer cells’ responses to CB1 and CB2 cannabinoids in their invasive and metastatic capacities.

### 3.4. CB1 Agonists AM-404, Arvanil, and Olvanil Induce Upregulation of the NF-kB Pathway

To determine the mechanism(s) of the induction of the EMT induced by the synthetic CB1 cannabinoids AM-404, arvanil, and olvanil, we investigated the effect of these CB1 agonists for their ability to upregulate nuclear factor-κB (NF-κB). Alaaeldin et al. [56] demonstrated that Azilsartan inhibited the NF-kB/IL-6/JAK2/STAT3 signaling pathway and inhibited the EMT expression in breast cancer cells. This concept elegantly supports the conceptual framework proposing the activation of the biased CB1 GPCR-Neu1-MMP9 signaling platform by a select group of cannabinoids activating the NF-kB pathway [6], implicated in enhanced EMT markers and metastasis.

To this end, we used RAW-Blue™ cells (mouse macrophage reporter cell line), which stably express a secreted embryonic alkaline phosphatase (SEAP) gene inducible by NF-kB and activator protein-1 (AP-1) transcription factors. AP-1, a transcription factor, regulates the expression of genes responding to various stimulus, including cytokines, growth factors, stress, and bacterial and viral infections. Upon stimulation, RAW-Blue™ cells activate NF-kB and AP-1, leading to the secretion of SEAP, which is detectable and measurable when using QUANTI-Blue™, a SEAP detection medium. The data in Figure 5 clearly demonstrate that CB1 agonist-treated cells with the inhibitors BIM-23, MMP-9i, and OP significantly inhibited SEAP secretion and, therefore, impacted NF-kB activity when compared to treatment with the agonist alone.

### 3.5. Synthetic CB1 Cannabinoids AM-404, Arvanil, and Olvanil Enhance the Migratory and Invasion Potential of Pancreatic PANC-1 Cancer Cells in a Scratch Wound Assay

The data depicted in Figure 4 clearly demonstrate that the synthetic CB1 cannabinoids AM-404, arvanil, and olvanil induce an invasive EMT N-cadherin expression on PANC-1 cells following 24 h exposure. These observations suggest that the effects of these synthetic CB1 cannabinoids may also influence the migratory potential of these cancer cells. The data depicted in Figure 1 also demonstrate that these synthetic CB1 cannabinoids can stage migratory intercellular communication by inducing tunneling nanotubes (TNTs) in cell cultures with the ability to induce spatial restriction and a communication specificity for invasion.

To confirm this concept, we investigated the migratory invasiveness of PANC-1 and SW-620 cancer cells treated with the synthetic CB1 cannabinoids AM-404, arvanil, and olvanil in a scratch wound assay. The data in Figure 6 depict the migratory rate of these CB1 cannabinoids from the scratch wound area over 12 h. The untreated PANC-1 control cells showed a near-complete wound closure rate of 5.97 ± 0.55 µm/h within 12 h (Figure 6A). In contrast, the wound closure rate for the CB1 cannabinoid-treated PANC-1 cells for arvanil was 9.93 ± 0.61 µm/h, for AM-404, it was 6.36 ± 0.63 µm/h, and for olvanil, it was 9.66 ± 0.5 µm/h, all occurring within 12 h. These data suggest that CB1 cannabinoids significantly induced the migration and invasiveness of PANC-1 cancer cells by potentially enhancing the epithelial−mesenchymal transition (EMT) for metastasis (Figure 6A).

In contrast, the untreated SW-620 control cells showed a near-complete wound closure rate of 19.65 ± 1.01 µm/h within 12 h (Figure 6B), which is 3.29 times faster than the wound closure rate for PANC-1. Interestingly, the wound closure rate for CB1 cannabinoid-treated SW-620 cells for arvanil was 15.02 ± 1.92 µm/h, for AM-404, it was 17.74 ± 1.54 µm/h, and for olvanil, it was 27.13 ± 2.4 µm/h, all occurring within 12 h (Figure 6B). These data suggest that CB1 cannabinoids had a minimal effect on the migration and invasiveness of SW-620 cancer cells, as these cells had already undergone the EMT process for invasiveness. Interestingly, olvanil-treated SW-620 cells significantly enhanced the migration and invasiveness of these cells (Figure 6B).

## 4. Discussion

The metabolism and cellular activities of cancer cells have been shown to be integrated and mutually regulated by tunneling nanotube communications. The in vitro effects of the synthetic CB1 cannabinoids AM-404, arvanil, and olvanil on human pancreatic PANC-1 and colorectal SW-620 cancer cell lines significantly altered cancer cells in forming missile-like shapes to induce tunneling nanotubes (TNT) communications in PANC-1 cells. Oseltamivir phosphate (OP) significantly prevented TNT formation. To understand the key metabolic driver pathways critical for pancreatic cancer progression, we used the AlamarBlue assay to determine synthetic CB1 cannabinoids to induce the cell’s metabolic viability drivers to stage migratory intercellular communication. The synthetic CB1 cannabinoids significantly increased cell viability compared to the untreated control for PANC-1 and SW-620 cells, and this response was significantly reduced with the neuromedin B inhibitor BIM-23127, neuraminidase-1 inhibitor OP, and MMP-9 inhibitor (MMP-9i). These synthetic CB1 cannabinoids also significantly increased N-cadherin and decreased E-cadherin EMT markers compared to the untreated controls, thereby inducing the process of metastatic phenotype for invasion. To confirm this concept, we investigated the migratory invasiveness of PANC-1 and SW-620 cancer cells treated with the synthetic CB1 cannabinoids AM-404, arvanil, and olvanil in a scratch wound assay. The synthetic CB1 cannabinoids significantly increased the rate of migration and invasiveness of PANC-1 cancer cells, whereas they had a minimal effect on the rate of migration of already metastatic SW-620 cancer cells. In contrast, olvanil-treated SW-620 cells significantly enhanced the migration rate and invasiveness of these cells. The data support the cellular and molecular mechanisms of the synthetic CB1 cannabinoids to orchestrate intercellular conduits to enhance cancer metabolic drivers to stage migratory intercellular communication in pancreatic cancer cells. In a recent report, Bunsick et al. [6] provided evidence that synthetic the CB1 agonists AM-404, arvanil, and olvanil each significantly and dose-dependently induced Neu-1 sialidase activity in live RAW-Blue macrophages, PANC-1, and SW-620 cells in vitro. Using the specific inhibitors BIM-23127, MMP-9i, and OP, CB1 agonist-induced sialidase activity was significantly blocked. These data support a novel signaling platform involving CB1 GPCRs and the crosstalk between Neu-1 and MMP-9 in receptor activation through the modification of the receptor glycosylation state. Using a co-localization assay, CB1 GPCRs co-localized in close proximity to Neu-1 on the cell sur-face, highlighting the potential interaction between CB1 GPCR and Neu1 [6].

Our group has previously found that the activation of Neu-1 sialidase results in the upregulation of the NF-kB pathway [57,58]. Since previous results have found that the CB1R and Neu-1 interact, we postulated that CB1 agonists activate Neu-1, activating the NF-kB pathway [6]. Our study results demonstrate that CB1 agonists induce NFκB-dependent secretory alkaline phosphatase (SEAP) activity by inducing the expression of epithelial–mesenchymal markers, E-cadherin, and vimentin in SW-620 cells compared to the untreated control. Upregulation of the NF-kB is also associated with the activation of inflammatory pathways (TNF-α), cell survival, and proliferation pathways, such as FLIP (FLICE-inhibitory protein) and CDKs, matrix remodeling, and angiogenic pathways, including MMPs and VEGF [59,60,61,62]. Additionally, NF-kB activation has been associated with the upregulation of epigenetic factors, including DNMT1 and histone modifications [62,63,64,65,66]. More research should be conducted on which synthetic cannabinoids are upregulating epigenetic markers to gain further understanding of how CB1 activation modulates cancer metastasis and cell survival.

To further substantiate our findings in our previous study [6], we hypothesized that CB1 agonists would affect the expression of EMT E-cadherin and N-cadherin markers in PANC-1 cells. We found that all three synthetic CB1 cannabinoids significantly reduced E-cadherin expression and increased N-cadherin (Figure 4). These results suggest that olvanil has a more significant impact on metastasis in PANC-1 cells than on SW-620 cells, as reported by Bunsick et al. [6]. For example, although all three CB1 agonists upregulated vimentin expression in SW-620 cells, olvanil had a weaker and non-significant effect on vimentin expression. The novelty of these findings signifies that CB1 agonist strength may impact the expression of vimentin, suggesting a kinetic component within the CB1 receptor that needs to be studied further. To explain these results, there are different cannabinoid allosteric ligands with different degrees of modulation called ‘biased modulation’ that can vary dramatically in a probe- and pathway-specific manner. The response seen here may be a result of specific CB1 functional selectivity, not from differences in the orthosteric ligand efficacy or stimulus–response coupling. Another interesting result is that arvanil had a more substantial effect on vimentin expression (SW-620) [6] than on N-cadherin expression (PANC-1), which may indicate that arvanil has a weaker impact on PANC-1 cells than on SW-620 cells.

E-cadherin up- or downregulation and its topography may have different effects on cells with a more epithelial or mesenchymal phenotype. This concept is a very important point in considering that these cancer cells, having a hybrid phenotype in relation to EMT markers, can invade using two different mechanisms involving either a single or collective cell invasion. For example, the synthetic CB1 cannabinoids significantly increased the rate of the scratch migration of PANC-1 cancer cells, whereas they had minimal effect on the rate of migration of already metastatic SW-620 cancer cells. In contrast, olvanil-treated SW-620 cells significantly enhanced the scratch wound migration rate of these cells. Bunsick et al. [6] reported that SW-620 cells showed no statistically significant reduction in E-cadherin following treatment with the CB1 cannabinoids AM-404, arvanil, and olvanil when compared to the control cohort, suggesting that the SW-620 cells had already undergone the EMT process for invasiveness. The SW620 cell line originated from a colon cancer lymph node metastasis (Duke’s type C), and they had a rounded or spindle-shaped fibroblast-like morphology [48]. Maamer-Azzabi et al. [67] reported that the expression of E-cadherin was slightly diminished in SW620 cells compared to another cell line of SW480 cells. Cytokeratin 18, which is another epithelial marker, was greatly increased in the SW620 cells. In contrast, for PANC-1 cells, CB1 cannabinoids significantly decreased E-cadherins and concomitantly increased N-cadherins compared to the untreated control. The data depicted in Figure 1 suggest that the synthetic CB1 cannabinoids AM-404, arvanil, and olvanil can significantly alter the cancer cell phenotype to induce tunneling nanotubes (TNTs) in PANC-1 cells and thereby orchestrate intercellular conduits to stage migratory intercellular communication. Gradiz et al. [68] reported that PANC-1 cells do not express E-cadherin but have epithelial phenotype characteristics since they express CK5.6 AE1/AE3 and mesenchymal vimentin. The data in Figure 4 illustrate that under our culture conditions, PANC-1 cells expressed E-cadherin, which was significantly decreased following synthetic CB1 agonists. Loh et al. [50] reported on the complexity of the cadherin switch in cancer cells. The upregulation of N-cadherin followed by the downregulation of E-cadherin identifies this cadherin switch, and this process is regulated by a complex network of signaling pathways and transcription factors.

Metastasizing cancer cells can surreptitiously adapt to their metabolic activity during their invasion by inducing EMT markers. Cancer cells can initiate their communications for invasion to reprogram their cellular activities. During this reprogramming process, cancer cells’ metabolism and other cellular activities are integrated and mutually regulated by tunneling nanotube communications to alter their specific metabolic functional drivers of tumor growth and progression. One critical aspect of cancer progression is cell viability. Understanding cell viability is crucial in cancer research as it impacts cell proliferation, growth, and metastasis and helps determine the efficacy of potential therapeutics [69,70]. The AlamarBlue assay found that all three synthetic cannabinoids increased cell viability in PANC-1 and SW-620 cell lines. An important finding in these experiments is that BIM-23, MMP-9i, and OP reduced cell viability compared to the untreated controls. These findings are significant as they indicate that the presence of these synthetic cannabinoids is increasing the viability of cancer cells and that the inhibition of the Neu-1 signaling paradigm reduces viability despite the presence of these synthetic cannabinoids. Moreover, it indicates that these synthetic cannabinoids increase their viability, metabolic functional drivers, and communications for invasion through Neu-1 activation.

An essential step in metastasis is cell migration, which is the movement of cells from their primary cancer location to other parts of the body. Several factors impact cell migration, including EMT and cell adhesion [71,72]. Since the synthetic cannabinoids have increased EMT and cell metabolic viability drivers in PANC-1 and SW-620 cells, we hypothesized that these synthetic cannabinoids could increase the migration rate of these cells. To determine this, we performed a scratch wound assay. Here, the synthetic cannabinoids increased cell migration in PANC-1 cells, but only olvanil increased migration in SW-620 cells. To explain these results, the data shown in Figure 5 may indicate that the migration rate was increased for the first 6 h compared to the untreated control. However, after six hours, the migration rate plateaued. It is possible that the cells depleted the synthetic cannabinoid quickly, leaving a limited amount available, thereby decreasing migration.

We aimed to determine the impact these synthetic cannabinoids have on cell shape and cellular projections. After treating PANC-1 cells with synthetic cannabinoids for 24 h and adding a CellMask membrane stain, we found that the cells significantly increased cell projections compared to the untreated control. The same experiments were conducted on the SW-620 cell line. However, the size of the cell and the zoom of the microscope were too small to determine any changes to the morphology and any cellular projections. As a result, no images were included. Tunneling nanotubes (TNTs) are based on their size and shape, and these membranous tubes interconnect and facilitate intercellular communication between cells. TNTs can exchange cellular components such as proteins and organelles and genetic material from miRNAs. As a result, TNTs facilitate regeneration, disease progression, immune responses, and cell signaling that modulates proliferation, migration, and apoptosis between cells [73,74,75,76]. If these nanotubes are present in PANC-1 cells following synthetic cannabinoid treatment, it may explain why these cells are undergoing EMT and increasing viability. Although each synthetic cannabinoid increased cell projections, arvanil had the weakest effect on PANC-1 cells. Based on previous experiments, it is likely that arvanil has a weaker effect on PANC-1 cells than SW-620 cells, which may be occurring due to functional selectivity.

Additionally, the phenotype of the cells changed following the treatment of the CB1 agonists compared to the untreated control. The control cells appear to be rounder, whereas the treated cells have a flatter, more ‘jagged’ appearance with more missile-like cellular projections. PANC-1 cells have an epithelial phenotype, and as epithelial cells become more metastatic, they lose their round appearance and become flatter and larger due to the loss of cadherins. A report by Dent et al. [77] found that cells with a more protrusive phenotype penetrate surrounding tissues and migrate to distant sites, thereby enhancing metastasis. Following treatment with OP, projection formation was significantly reduced, the cell size was reduced, and the shape was more representative of the untreated control, suggesting that OP treatment inhibits cell projections and reduces cell size.

## 5. Conclusions

Understanding the role of CB1 GPCRs on cancer metastasis and progression is difficult due to conflicting evidence that it enhances and prevents progression [57,78]. However, the functional selectivity of CB1 GPCRs is likely why we are seeing a lack of a clear consensus when it comes to cannabinoid treatments. Our previous study provided evidence that CB1 receptors have a potential interaction with Neu-1 through heterodimerization with NMBR GPCRs to activate MMP9 and Neu-1 and induce glycosylated receptors and downstream signaling [6]. Here, we used synthetic CB1 cannabinoids and found them to upregulate cancer cell viability, EMT markers, and migration involving this Neu-1 signaling paradigm. However, these synthetic CB1 cannabinoids had a variable impact on cancer cell lines, eliciting a more substantial effect in one cell line but a weaker effect in another (arvanil and olvanil). The variability of potency through different cell lines is likely indicative of functional selectivity as each agonist may elicit a different effect due to different binding conformations of the cell, or the conditions present in each cell line may impact its effect. Moreover, a more comprehensive biochemical analysis should be performed on this signaling paradigm to validate these results further.

The findings of this study and the previous one [6] lead us to conclude that synthetic CB1 cannabinoids may elicit epigenetic changes in these cancer cells because the NF-kB pathway influences many histones and DNA methylation modifications [58,59,60,61,62]. Based on recent reports, there are promising epigenetic markers to investigate, such as the trimethylation marker of H3K27me3 and H3K4me3 [62], miR-10b, miR-17, miR-21, and miR-9, all of which are affected by changes in NF-kB expression [63,64,65,66]. Understanding how these epigenetic changes are elicited through CB1 cannabinoid activation may provide novel avenues for therapeutics, biomarkers, and understanding of how these cancers progress.

## Figures and Tables

**Figure 1 cells-14-00071-f001:**
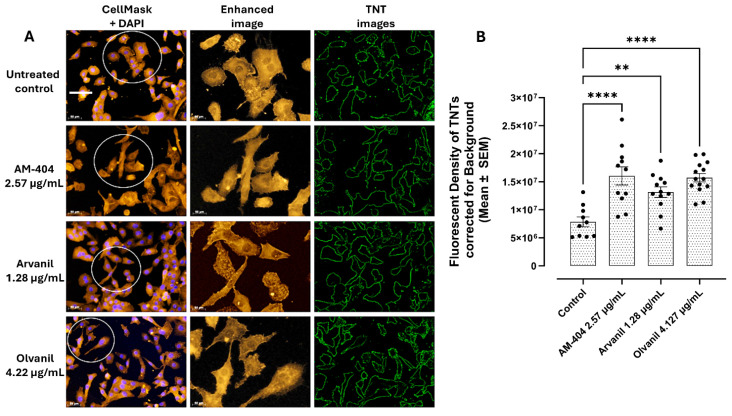
Synthetic CB1 cannabinoids AM-404, arvanil, and olvanil significantly alter the cancer cell phenotype to induce tunneling nanotubes (TNTs) in PANC-1 cells. (**A**) Representative CellMask + DAPI and (**B**) TNT quantification images of cell size and projections following synthetic cannabinoid treatment. Slide images were observed using Zeiss M2 epi-fluorescent microscopy (20× magnification, scale bar), capturing images under the Rhodamine (554 nm) channel. The images were enhanced (TNT images), and the TNT image cell projections were differentiated using ImageJ. The projections were quantified using Corel Photo-Paint. (**C**) Representative TNT images of cells treated with oseltamivir phosphate and (**D**–**F**) quantification of TNTs of PANC-1 cell treatment with 300 µg/mL oseltamivir phosphate (OP) for 15 min before synthetic cannabinoid treatment. **** *p* < 0.0001, *** *p* < 0.001, ** *p* < 0.01.

**Figure 2 cells-14-00071-f002:**
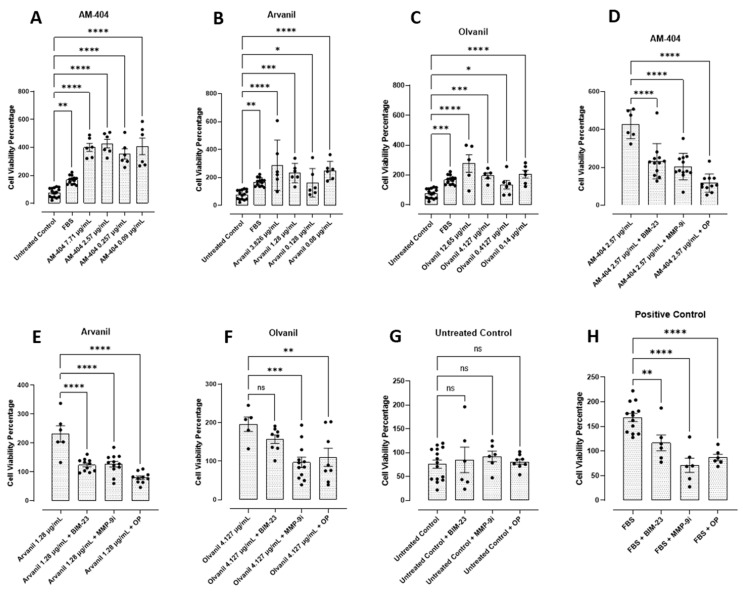
Synthetic CB1 cannabinoids AM-404, arvanil, and olvanil increase cell viability (**A**–**D**) and metabolic activity in PANC-1 cells using the AlamarBlue assay. PANC-1 cells were plated with ~20,000 cells per well in a flat-bottom 96-well plate containing conditioned culture–medium for 24 h at 37 °C and 5% CO_2_. Following incubation, the cells were treated with 90 μL of each CB1 agonist at the indicated doses in a medium without FBS for 24 h alone or in combination with the BIM23 (20.9 nM), MMP9i (5 nM), or OP (300 µg/mL) inhibitors (**E**–**G**) for 15 min before ligand stimulation. (**G**,**H**) Untreated controls with inhibitors or inhibitors with FBS only. A total of 10 μL of AlamarBlue reagent was added to each well and incubated for 4 h. The absorbance was recorded using a spectrophotometer at 570 nm. Data are represented as the mean ± SEM of 3 independent experiments performed in triplicate. Cell viability was determined based on the fluorescence corrected for blank media control and untreated cellular control. As indicated by asterisks, statistical significance was calculated with ANOVA and Fisher’s uncorrected LSD post hoc test at a confidence level of 95%. ns = non-significant, **** *p* < 0.0001, *** *p* < 0.001, ** *p* < 0.01, and * *p* < 0.05.

**Figure 3 cells-14-00071-f003:**
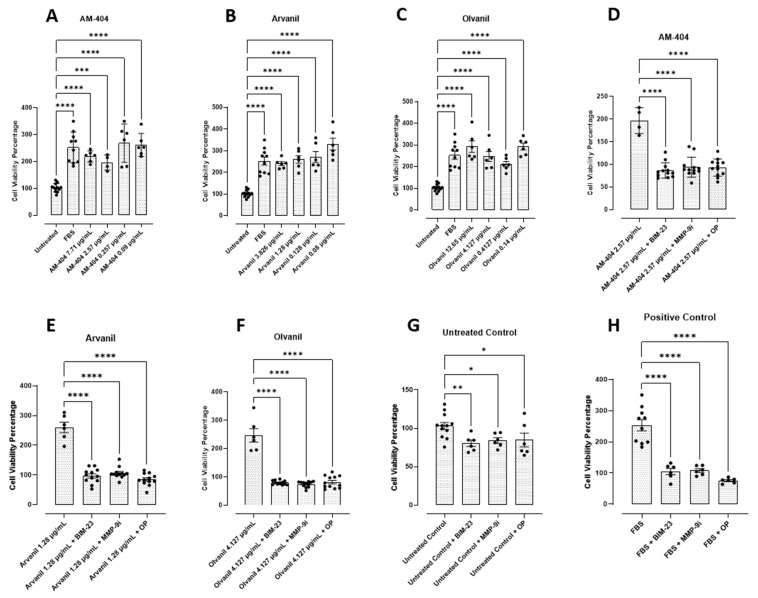
Synthetic CB1 cannabinoids AM-404, arvanil, and olvanil increase cell viability (**A**–**C**) and metabolic activity in SW-620 cells using the AlamarBlue assay. SW-620 cells were plated with ~20,000 cells per well in a flat-bottom 96-well plate containing conditioned culture–medium for 24 h at 37 °C and 5% CO_2_. Following incubation, the cells were treated with 90 μL of each CB1 agonist at the indicated doses in a medium without FBS for 24 h alone or in combination with the BIM23 (20.9 nM), MMP9i (5 nM), or OP (300 µg/mL) inhibitors (**D**–**F**) for 15 min before ligand stimulation. (**G**,**H**) Untreated controls with inhibitors or inhibitors with FBS only. A total of 10 μL of AlamarBlue reagent was added to each well and incubated for 4 h. The absorbance was recorded using a spectrophotometer at 570 nm. Data are represented as the mean ± SEM of 3 independent experiments performed in triplicate. Cell viability was determined based on the fluorescence corrected for blank media control and untreated cellular control. As indicated by asterisks, statistical significance was calculated with ANOVA and Fisher’s uncorrected LSD post hoc test at a confidence level of 95%. ns = non-significant, **** *p* < 0.0001, *** *p* < 0.001, ** *p* < 0.01, and * *p* < 0.05.

**Figure 4 cells-14-00071-f004:**
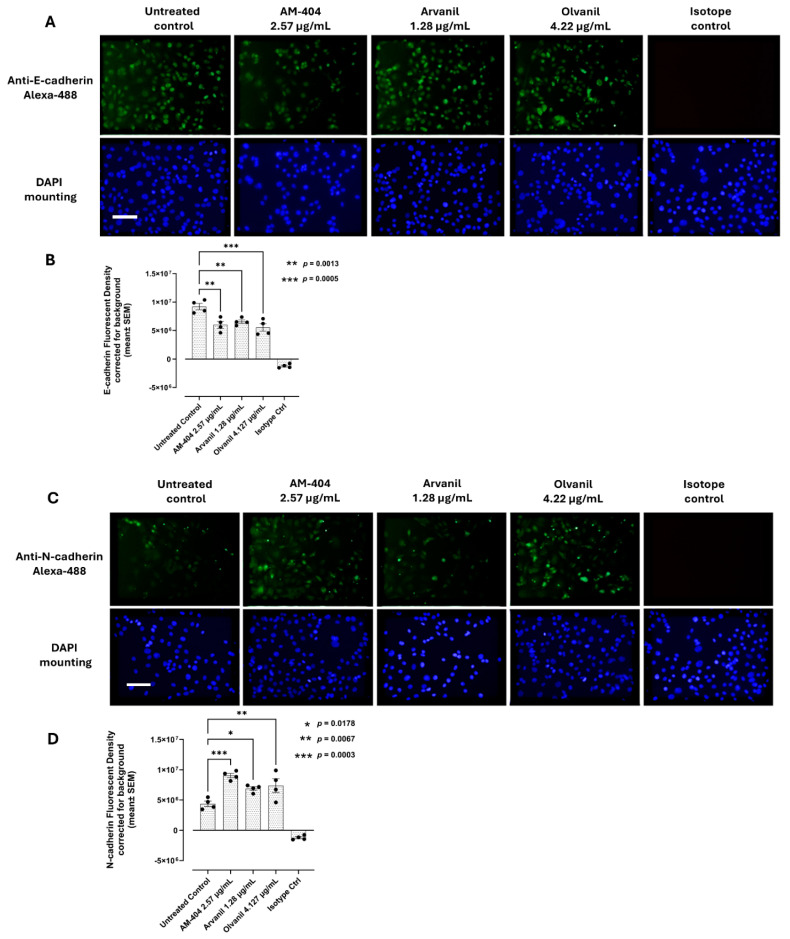
Synthetic cannabinoids AM-404, arvanil, and olvanil concomitantly decrease E-cadherin (**A**,**B**) and increase N-cadherin (**C**,**D**) in PANC-1 cells. Immunofluorescence analysis of E- and N-cadherin expressions in response to CB1 cannabinoids. CB1 agonists AM-404, arvanil, and olvanil significantly reduced the expression of the epithelial–mesenchymal transition (EMT) marker E-Cadherin with concomitant increase in N-cadherin in PANC-1 cells. Cells were fixed, permeabilized, blocked, and immunostained with primary mouse monoclonal IgG antibodies for E- or N-cadherin, followed by goat anti-mouse Alexafluor 488 (green) secondary antibodies. As a negative isotope control, we used normal mouse IgG. Images were captured using Zeiss Imager M2 epi-fluorescent microscopy (20× magnification, scale bar). The eight fluorescence images were analyzed using Corel Photo-Paint, with an average of eight points subtracted by the background fluorescence and multiplied by the pixel density. These data were representative and reproducible in three independent experiments. As indicated by asterisks, statistical significance was calculated with ANOVA and Fisher’s uncorrected LSD post hoc test at a confidence level of 95%. * *p* values indicated in the figure.

**Figure 5 cells-14-00071-f005:**
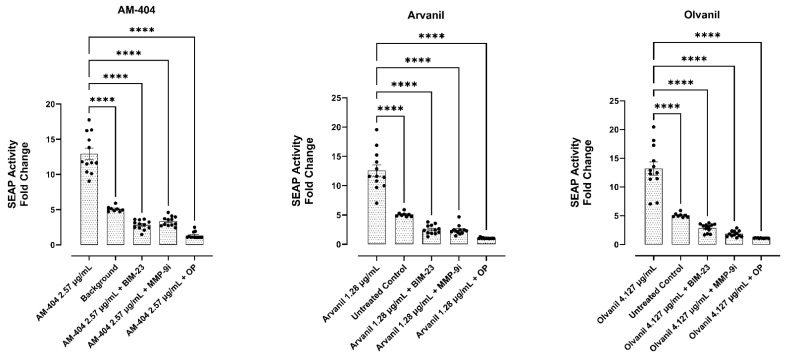
Synthetic CB1 cannabinoids AM-404, arvanil, and olvanil increase NF-kB expression and are inhibited by Neu-1 inhibitors BIM23, MMP9i, and OP. RAW-Blue cells stably express a secreted embryonic alkaline phosphatase (SEAP) gene inducible by NF-kB and AP-1 transcription factors. Upon stimulation, RAW-Blue™ cells activate NF-kB and AP-1, leading to the secretion of SEAP, which is detectable and measurable when using QUANTI-Blue™, a SEAP detection medium. Quantitative spectrophotometry analysis of the effect of AM-404, arvanil, and olvanil-induced SEAP activity in the culture medium was made. The measurement of the relative SEAP activity was calculated as fold change in each compound (SEAP activity in medium from treated cells minus medium divided by SEAP activity in medium from untreated cells minus background). Results are the means of three separate experiments. As indicated by asterisks, statistical significance was calculated with ANOVA and Fisher’s uncorrected LSD post hoc test at a confidence level of 95%. **** *p* < 0.0001. Data were collected over 3 independent experiments (n = 3).

**Figure 6 cells-14-00071-f006:**
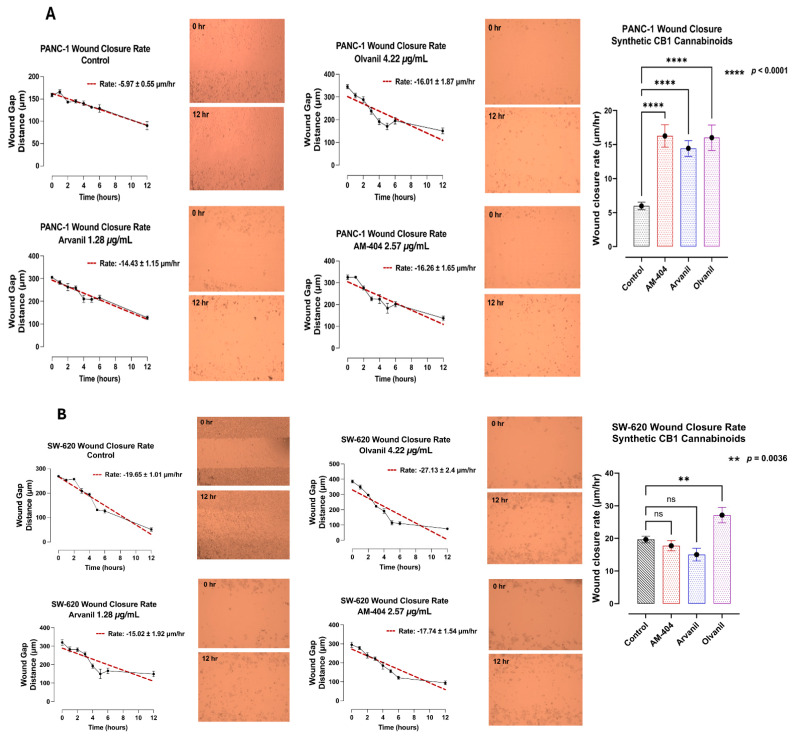
Synthetic CB1 cannabinoids AM-404, arvanil, and olvanil affect the migratory and invasion potential of (**A**) pancreatic PANC-1 and (**B**) colorectal SW-620 cancer cells in a scratch wound assay. PANC-1 and SW-620 cells were cultured in a 50 mm glass bottom MatTek culture dish and allowed to adhere to 90% confluence on the glass slide in an incubator at 37 °C and 5% CO_2_. A sterile pipette tip created a scratch wound, and non-adherent cells were removed after washing. The control cells were supplemented with a medium containing 5% fetal bovine serum (FBS). In contrast, the treated cells were supplemented with CB1 cannabinoids AM-404, arvanil, or olvanil. Imaging was taken with a Nikon Eclipse Ti2 microscope (4× magnification) every hour for the first 6 h and at 12 h after the creation of the scratch wound. The wound width was measured at 6–8 points per image using the microscope NIS-Elements AR software, version 5.21.00, and the results were analyzed to create a simple linear regression (red dashed line). GraphPad Prism 10 was used to measure the rate of wound closure represented by μm/h. The slope of the linear regression line is the rate of wound gap closure in μm/h ± standard error of the slope. The quantified data represent two to three independent experiments displaying similar results. Statistical significance, as indicated by asterisks, was calculated with ANOVA and Fisher’s LSD uncorrected multiple comparisons at a confidence level of 95%. ns = non-significant, **** *p* < 0.0001, and ** *p* < 0.0036.

## Data Availability

All data needed to evaluate the paper’s conclusions are present. The preclinical data sets generated and analyzed during the current study are not publicly available but are available from the corresponding author upon reasonable request. The data will be provided following the review and approval of a research proposal, statistical analysis plan, and execution of a data sharing agreement. The data will be accessible for twelve months for approved requests, considering possible extensions; contact szewczuk@queensu.ca for more information on the process or to submit a request.

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
