# Peer review of "Synthetic CB1 Cannabinoids Promote Tunneling Nanotube Communication, Cellular Migration, and Epithelial–Mesenchymal Transition in Pancreatic PANC-1 and Colorectal SW-620 Cancer Cell Lines"

_cells, 2025, doi:10.3390/cells14020071_

Round 1

Reviewer 1 Report

Comments and Suggestions for Authors

The manuscript submitted by the authors focuses on the effect of synthetic canabinoids on
malignant tumor cell lines (colorectal cancer and pancreatic cancer). However, the results obtained
from the experimental studies do not support the conclusions drawn by the authors. In the
introduction and discussion, the authors describe that canabinoids act via the canabinoid receptor
and that G-protein-coupled (GPCR) signaling is involved in the development of the canabinoid effect.
However, no mechanistic investigations are carried out in this regard. Although the NF-kB signaling
pathway and cell viability are addressed by certain inhibitors in order to test the involvement of NF-
kB and the modulation of cell viability, the elaborated results remain independent and were not
investigated in the context of CB1 receptor activation or GPCR-coupled signaling. The same applies to
the other pancreatic carcinoma-associated biomarkers under study, i.e. (i) tuneling nanotubles (TNT),
(ii) metabolic activity, (iii) migratory capacity, and (iv) the EMT markers under study, i.e. vimentin and
N-cadherin. Due to the lack of mechanistic experiments showing a causal relationship between the
modulation of the above-mentioned biomarkers and CB1 receptor activation and/or GPCR signaling,
the results are purely descriptive, i.e. cause-effect principles are not discernible.
One example of how the conclusions drawn from the results of the experiments are not
tenable is that the wound scratch assay only reveals a migratory potential, but not the ability of
tumor cells or cells per se to invade or even metastasize. Migration is a cell behavior that can occur
completely independently of tumor cell invasion or metastasis, e.g. wound healing. The authors also
do not investigate any veritable markers for invasion or metastasis, such as certain matrix-degrading
proteases, which have already been published by several groups with respect of a poor prognosis in
the context of pancreatic cancer. However, an increase in the metastatic potential normally requires
in vivo testing in appropriate animal experiments.
The cell lines chosen by the authors are actually unsuitable for confirming the
biomarkers/features investigated as cancer drivers, as they were established from malignant tumor
tissue and therefore represent the final stage of the multistep process of carcinogenesis. Normal
pancreatic cell lines would be helpful to investigate the role of the actually tested biomarkers in
pancreatic carcinogenesis. Such cell lines are currently available.
A serious factual error in the manuscript is that the authors write in the abstract: "CB1
cannabinoids also significantly increased N-cadherin and decreased E-cadherin EMT markers
compared to the untreated controls, inducing the process of metastatic phenotype for invasion“.
Objectively speaking, it is not the case that the induction of the metastatic phenotype is the
prerequisite for invasion, but rather the reverse: without invasion, metastasis cannot take place.
Against the background of these experimental and factual weaknesses, it is difficult in
principle to consider the present manuscript at all, since a revision would require a complete
restructuring of the experiments. I therefore recommend that the Editorial Board reject the
manuscript in its current form.

Author Response

Comments and Suggestions for Authors

I have no comments and/or suggestions for authors.

Submission Date

18 September 2024

Date of this review

03 Oct 2024 16:58:36

Author response: Thank you for the support.

Reviewer 2 Report

Comments and Suggestions for Authors

In this manuscript, Bunsick DA et al. showed that synthetic CB1 cannabinoids regulate tunneling nanotube (TNT) communications and enhance cell viability and promote scratch wound migration in pancreatic PANC-1 and colorectal SW-620 cancer cells. Oseltamivir Phosphate(Tamiflu), which is a potent and selective inhibitor of the neuraminidase, significantly prevented TNT formation and NMBR inhibitor, BIM-23127 and MMP-9 inhibitor (MMP-9i) abrogated synthetic CB1 cannabinoids’ effects. Manuscript is well written and data is interesting, although several points need to be improved.

Major points.

1 They suggest that the effects of CB1 agonists are through NFkB signaling by NFkB-dependent secretory alkaline phosphatase (SEAP) activity (Fig. 5), but this experiments alone is insufficient. They should show CB1-dependent NFkB nuclear localizaton by immunofluorescence.

2 In Fig. 4 and 6, are these CB1 agonists-dependent EMT regulation and wound migration inhibited by the inhibitors used in other experiments ? If possible, they should perform experiments with these inhibitors.

Minor points.

3 The order of Figure 1 should be corrected. In general, A is upper side. And They do not describe E and F.

4 Figure legends of Fig. 2 and 3 are insufficient, they do not describe A-H. Correct them.

Author Response

Comments and Suggestions for Authors

In this manuscript, Bunsick DA et al. showed that synthetic CB1 cannabinoids regulate tunneling nanotube (TNT) communications and enhance cell viability and promote scratch wound migration in pancreatic PANC-1 and colorectal SW-620 cancer cells. Oseltamivir Phosphate (Tamiflu), which is a potent and selective inhibitor of the neuraminidase, significantly prevented TNT formation and NMBR inhibitor, BIM-23127 and MMP-9 inhibitor (MMP-9i) abrogated synthetic CB1 cannabinoids’ effects. Manuscript is well written and data is interesting, although several points need to be improved.

Author response: Thank you for the comment and support.

Major points.

1 They suggest that the effects of CB1 agonists are through NFkB signaling by NFkB-dependent secretory alkaline phosphatase (SEAP) activity (Fig. 5), but this experiments alone is insufficient. They should show CB1-dependent NFkB nuclear localizaton by immunofluorescence.

 Author response: Thank you for the comment.

          This immunofluorescence technique, as recommended, is quite challenging and can present significant bias, such as reaction bias and interruption bias [Biomark Insights 2010, 5, 9–20]. Reaction bias is described as pitfalls at the technical level, such as specimen fixation, tissue processing, detection system, and tissue pretreatment. Similarly, pitfalls in the design and interpretation level are interpretation bias, which includes the specificity and sensitivity of the antibodies. In addition, this technique has the possibility of having a high background and species cross-reactivity. Also, there are isotype subunits of NFkB: NF-κB subunits p50, p65, and p75.

            The NF-kB complex is sequestered in a dimer form (with p65/p50 dimers as the most common composition) in the cytoplasm, where IkB binds it. Upon activation by various stimuli, IKK interacts with the inhibitory IkB, resulting in phosphorylation, ubiquitination, and degradation of IkB, rendering the dimer free to translocate to the nucleus. Here, the dimer binds to kappa B binding sites of several gene targets that may be involved in cell survival in cells or inflammatory pathways. Dimers consisting of p65/p50 tend to be transcriptionally active, whereas p50 homodimers tend to suppress gene transcription.

          NFkB signaling by NFkB-dependent secretory alkaline phosphatase (SEAP) activity involves the complex signaling pathways as described above in addition to the activation of AP-1. Upon stimulation, RAW-Blue™ cells

activate NF-κB and AP-1, leading to SEAP secretion, detectable and measurable using QUANTI-Blue™ (InvivoGen) SEAP in the medium. RAW-Blue™ cells are made to be resistant to Zeocin™ and G418 antibiotics.

          We have previously published an extensive peer-reviewed article on CB1 agonists using the     NFkB signaling by NFkB-dependent secretory alkaline phosphatase (SEAP) without doing NFkB nuclear localization by immunofluorescence (Bunsick et al. Cells 2024, 13, 480. https://doi.org/10.3390/cells13060480).

2 In Fig. 4 and 6, are these CB1 agonists-dependent EMT regulation and wound migration inhibited by the inhibitors used in other experiments? If possible, they should perform experiments with these inhibitors.

 Author response: Thank you for the comment. Your suggestion to use inhibitors BIM-23, MMP-9i, and OP in wound migration is excellent. Here, we wanted to investigate the efficacy of these CB1 agonists in promoting the migration potential of cancer cells. Interestingly, we found a differential effect of the CB2 agonists on PANC-1 and SW620 cancer cells. CB1 cannabinoids had minimal effect on the migration and invasiveness of SW-620 cancer cells, which cells have already undergone the EMT process for invasiveness. Interestingly, Olvanil-treated SW-620 cells significantly enhanced the migration and invasiveness of these cells. On the other hand, CB1 cannabinoids significantly induced the migration and invasiveness of PANC-1 cancer cells by potentially enhancing the epithelial−mesenchymal transition (EMT) for metastasis.

          The use of inhibitors in these experiments requires initial data to elucidate the potential of the CB1 agonists to induce the migratory and invasion potentials of cancer cells in a scratch wound assay. These experiments are in progress in a follow-up manuscript.

Minor points.

3 The order of Figure 1 should be corrected. In general, A is upper side. And They do not describe E and F.

Author response: Thank you for the comment. We have corrected Figure 1. 

4 Figure legends of Fig. 2 and 3 are insufficient, they do not describe A-H. Correct them.

Author response: Thank you for the comment. We have corrected Figure 2and 3. 

Submission Date

18 September 2024

Date of this review

30 Sep 2024 15:02:32

Reviewer 3 Report

Comments and Suggestions for Authors

In this study the authors aimed at investigating the effect of CB1 cannabinoids, specifically AM-404, Arvanil, and Olvanil, on PANC-1 and SW-620 cells in terms of modulating tunneling nanotubes, cell viability and migration, and on some EMT markers.  

The manuscript is clearly written and the results clearly presented. However, some important issues influence the conclusions that, in my opinion, are not fully supported by the presented results.

Materials and Methods, paragraph 2.9 and relative results: for the scratch test the authors report that “The control cells were then supplemented with a medium containing 5% FBS. At the same time, the CB1 agonist-treated cells were supplemented with AM-404, Arvanil, or Olvanil in a medium without FBS”. The test should be performed using the dame FBS % (serum free) to have the same cell proliferation and to avoid the influence of cell proliferation on the result. If I understood correctly, control samples were cultured in presence of FBS. If so, the results on the migration are inconsistent.

Results, paragraph 3.3 and figure 4: the authors report an effect of the treatment on the expression of E-cadherin and N-cadherin. The micrographs in figure 4 are too small and some insets or higher magnification micrographs should be included. Moreover, an important point is that the text and the figure legend do not describe the pattern of expression and the localization of the two cadherin. Especially for E-cadherin, is fundamental the description of its localization, that is if the protein is expressed at the cell plasma membrane (cell junction functional) or in the cytoplasm (cell junction degraded and not functional).  The quantitative analysis is not sufficiently informative.

Moreover, figure 4 shows the immunofluorescence analysis of E-and N-cadherin expressions in response to CB1 cannabinoids in PANC-1 and the same results obtained on SW620 cells were published by the same authors (Cells. 2024 Mar; 13(6): 480). Please comment this and clearly highlight the novelty of the results compared to the already published paper.

Author Response

Comments and Suggestions for Authors

In this study the authors aimed at investigating the effect of CB1 cannabinoids, specifically AM-404, Arvanil, and Olvanil, on PANC-1 and SW-620 cells in terms of modulating tunneling nanotubes, cell viability and migration, and on some EMT markers. 

            The manuscript is clearly written and the results clearly presented. However, some important issues influence the conclusions that, in my opinion, are not fully supported by the presented results.

Author response: Thank you for the comment and support.

Materials and Methods, paragraph 2.9 and relative results: for the scratch test the authors report that “The control cells were then supplemented with a medium containing 5% FBS. At the same time, the CB1 agonist-treated cells were supplemented with AM-404, Arvanil, or Olvanil in a medium without FBS”. The test should be performed using the dame FBS % (serum free) to have the same cell proliferation and to avoid the influence of cell proliferation on the result. If I understood correctly, control samples were cultured in presence of FBS. If so, the results on the migration are inconsistent.

Author response: Thank you for the comment. We have corrected this text error. “The control cells were then supplemented with a medium containing 5% FBS. At the same time, the CB1 agonist-treated cells were supplemented with AM-404, Arvanil, or Olvanil in a medium with 5% FBS”.

Results, paragraph 3.3 and figure 4: the authors report an effect of the treatment on the expression of E-cadherin and N-cadherin. The micrographs in figure 4 are too small and some insets or higher magnification micrographs should be included. Moreover, an important point is that the text and the figure legend do not describe the pattern of expression and the localization of the two cadherin. Especially for E-cadherin, is fundamental the description of its localization, that is if the protein is expressed at the cell plasma membrane (cell junction functional) or in the cytoplasm (cell junction degraded and not functional).  The quantitative analysis is not sufficiently informative.

Author response: Thank you for the comment. According to the M&M protocol, “cells were media-starved and incubated with predetermined indicated concentrations of CB1 agonists (AM 404, Olvanil, and Arvanil) in the designated wells for 24 h. Control wells were incubated with media without FBS. Cells were then fixed at 4% PFA (300 µL) and incubated in the cold for 24 h. To facilitate the binding of antibodies, cells were permeabilized with 0.2% Triton x100 (300 µL) for 5 min and, then blocked with 4% BSA/Tween20/TBS (300 µL) for 24 h in the cold to prevent non-specific binding. Primary antibodies were added at a 1:10 dilution in 4% BSA/Tween20/TBS solution and incubated in the fridge for 24 hours. Subsequently, the secondary antibodies were added at a 1:1000 dilution in PBS and incubated in the fridge for 24 hours. Cells treated with secondary antibodies were employed as a negative control.”

          The protocol described here would provide E- and N-cadherin staining on the membrane as well as intracellular. We have followed the identical protocol of E- and N-cadherin staining as previously reported by us (Bunsick et al. Cells 2024, 13, 480. https://doi.org/10.3390/cells13060480 ).

Moreover, figure 4 shows the immunofluorescence analysis of E-and N-cadherin expressions in response to CB1 cannabinoids in PANC-1 and the same results obtained on SW620 cells were published by the same authors (Cells. 2024 Mar; 13(6): 480). Please comment this and clearly highlight the novelty of the results compared to the already published paper.

Author response: Thank you for the comment.

Submission Date

18 September 2024

Date of this review

30 Sep 2024 18:18:10

Reviewer 4 Report

Comments and Suggestions for Authors

The manuscript entitled “Synthetic CB1 Cannabinoids Surreptitiously Orchestrate Tunneling Nanotubes to Enhance Cancer Metabolic Drivers to Stage Migratory Intercellular Communication in Promoting Scratch Wound Migration and Epithelial–Mesenchymal Transition in Pancreatic PANC-1 and Colorectal SW-620 Cancer Cell Lines” focused on the response of cancer cell lines to synthetic cannabinoids.  The authors need to address several issues and make corrections to the overall writing and flow of the manuscript before it should be considered for publication. If this manuscript is revised and resubmitted, please provide line numbers to facilitate reviewers providing feedback on grammatical errors or inconsistencies throughout the document – without them, it will be too cumbersome to point these out throughout the entire document. 

I have major concerns with this manuscript:

1.  The introduction mentions that CB1 receptors are overly expressed in the CNS, yet there is no mention of the distribution of these receptors outside the CNS.  Why were these two cancer cell lines used for this study?  There is no data provided on the level of expression of CB1 receptors on these cells – do they express CB1 receptors or do they express other receptors that can bind to cannabinoids?  If this was in another study, then a discussion of this is needed in the introduction. If not, then the authors should provide data or other justification for using these cell lines to study the effects of cannabinoids.

2. Also in the introduction, paragraph 2 discusses processes that synthetic cannabinoids have been shown to affect. These seem to support that there are various other receptors that may be involved, in which the authors do not elaborate.

3.  The sentence “CB1, a primary target for synthetic cannabinoids, can cause diverse cellular responses depending on the agonist used [8].” Is vague – what is meant by this?  How many different agonists are there for CB1 receptors, and how variable are these responses?  It would also be nice to discuss the rationale for using the neuraminidase inhibitors chosen – is there any data that these affect cancer cells and/or signaling from CB1 receptors?

4. In the introduction, the authors jump from discussing NF-κB and API (AP-1?) signaling involving Neu-1 and MMP-9, to then discussing DNA methylation, and then back to NF-κB pathways, which made for a confusing introduction. After reading the introduction, I lost sight of the main point of the manuscript was to show data on cancer cell migration/EMT.  The authors do not show any data on signaling pathways or epigenetic regulation, yet the introduction leads the reader this direction Their last paragraph clearly states “Here, we investigated the effects of synthetic CB1 cannabinoids on EMT markers, cell viability, migration and tunneling nanotubes signaling pathways in PANC-1 and SW-620 cancer cell lines.”  

5.  All figure legends are written like a rehash of the methods sections and need to be edited.

6.   Figure 1: Figures 1A-B are at the bottom, and there is no mention of Figures 1D-F in the legend.  The TNT images are hard to decipher; what, specifically is being measured?  The nanotubes between cells are not evident (to me), nor is there any mention in the introduction that cancer cells produce TNTs, the significance of them, and why these cells would produce them in response to cannabinoids.  Figure 1A and 1C CellMask + DAPI images look identical, except enhanced for brightness. Are these images altered? 

7.  Did the authors perform the same experiments as shown in Figure 1 for SW-620 cells?  Why were these data not included?  If they were done and not significant, then that should be stated at the very least. If they were done previously, the results should be stated here and cited.

8.  Figure 1C: the IF image of the cells after OP treatment using CellMask are not very uniform. Are the cells still viable, or did the OP treatment result in a substantial number of dead cells?  They do not look very healthy, although hard to say if they are dead or have just lifted from the dish. Data should be provided to determine health of the cells to better interpret the data.

9.  Figure 1 regarding TNTs:  Section 2.6 in the methods does not describe the TNT assay other than to mention that CellMask was used to delineate the cell membrane, so the TNT images appear to be outlines of the plasma membrane. Is the total fluorescence of the TNT images being quantitated?  This may be over interpreting the effects of treatments as OP treatment reduced cell size, so would naturally reduce the fluorescence of the cell membrane. This is either not clear or misleading. What projections are being measured, exactly? And what does each data point in Figures 1B, D-F represent (what is the sample size/experimental number)?

10. Figures 2 and 3: The y-axes are “Cell Viability Percentage” yet most of the values are over 100%.  What does this mean?  How does cell viability go from 100% to 250% between untreated cells and cells with FBS (as shown in Figure 3A, for an example). Is there any difference between cells with FBS vs AM-404 in Figure 3A (and same question for other treatments)? There is no mention of why there is not a dose-dependent response (i.e., not consistent changes in viability depending on dose) for some of the figures (e.g., Figures 2B-C and Figures 3A and 3C).  Were all comparisons done just between the control and each treatment?  Also for these figures, in the methods it is stated that the experiments were done in triplicate, but the legends read “3 independent experiments were performed in triplicate” and the figures have 9 separate data points per treatment.  Did you not average the technical replicates or are you treating those as independent data points?  This may skew your results.

11. Figure 4: I don’t see much difference in expression of E- or N-cadherins in the IF images. Can better images that represent the data be provided? Did the authors look at vimentin expression (or any other EMT markers) that may show more convincing results?  

12. Figure 5: While the data show an increase in SEAP expression in macrophages (and why this cell line was used, other than for this particular protocol?), this could be due to upregulation of NF-κB or AP-1, among others, potentially.  A confirmatory assay to show evidence of specific NF-κB upregulation is needed.

13. Figure 6: the images of the scratch assays are hard to see, and it appears that some of the images are at higher magnification than others. How was the rate of migration measured in these images?  Normally it is based on rate of wound closure, yet in some of these images, some cells migrated quite rapidly and others didn’t, so there is not a clear “margin” on either side to measure width of the wound.  For example, in Figure 6B control at 12 hours, the wound looks more closed in the image than what is shown at 12 hours for Olvanil, yet the graph indicates that Olvanil has a higher wound closure rate. Having an outline of what was measured in the images may help, but I’d suggest re-evaluating your data to ensure the data are being reported accurately and the conclusions you make in the discussion support the data.

14. In the discussion, paragraph 4, the authors mention that these synthetic cannabinoids reduce E-cadherin and increase N-cadherin and vimentin. No data were shown regarding vimentin, and there are many reports that indicate E- and N-cadherin expression levels are not the only (or best) markers to demonstrate an EMT process. Additionally, drawing conclusions between PANC-1 cells where data are provided in this manuscript to SW-620 cells (which the data may have been reported in reference #3) and draw conclusions about the impact of Olvanil on EMT between these two cells does not allow the reader to determine if this conclusion is accurate.   The data provided in Figure 4 are not very convincing; I’d recommend providing protein expression data and/or additional EMT markers to support their conclusions.

Additional, minor things to address:

1. The use of the word “surreptitiously” is misleading (title and in the manuscript) – are the cancer cells being “secretive, avoiding notice or attention”? 

2. The title is a mouthful and hard to interpret – I’d suggest a rewrite to be more concise/clear.

Comments on the Quality of English Language

Please edit the manuscript for grammatical errors and consistency in writing throughout.  

Author Response

Comments and Suggestions for Authors

The manuscript entitled “Synthetic CB1 Cannabinoids Surreptitiously Orchestrate Tunneling Nanotubes to Enhance Cancer Metabolic Drivers to Stage Migratory Intercellular Communication in Promoting Scratch Wound Migration and Epithelial–Mesenchymal Transition in Pancreatic PANC-1 and Colorectal SW-620 Cancer Cell Lines” focused on the response of cancer cell lines to synthetic cannabinoids. The authors need to address several issues and make corrections to the overall writing and flow of the manuscript before it should be considered for publication. If this manuscript is revised and resubmitted, please provide line numbers to facilitate reviewers providing feedback on grammatical errors or inconsistencies throughout the document – without them, it will be too cumbersome to point these out throughout the entire document.

Author response: Thank you for the comment. We have highlighted the changes in red text and yellow highlighted.

I have major concerns with this manuscript:

  1. The introduction mentions that CB1 receptors are overly expressed in the CNS, yet there is no mention of the distribution of these receptors outside the CNS. Why were these two cancer cell lines used for this study? There is no data provided on the level of expression of CB1 receptors on these cells – do they express CB1 receptors or do they express other receptors that can bind to cannabinoids? If this was in another study, then a discussion of this is needed in the introduction. If not, then the authors should provide data or other justification for using these cell lines to study the effects of cannabinoids.

Author response: Thank you for the comment. We have added the following in the introduction’s first paragraph:  “The CB1 GPCR receptors are also expressed in the periphery but at much lower levels than in the central nervous system (CNS) [3]. Recent studies indicated that CB1 GPCR receptors are expressed in pancreatic [4] and colorectal [5] cancer cells. Bunsick et al. [6] have reported that CB1 agonists can influence both pancreatic and colorectal cancer cellular behavior by modulating GPCR signaling pathways via a functional selectivity mechanism(s), which were found to be essential in regulating cell viability, migration, and epithelial-mesenchymal transition (EMT). Cherkasova et al. [5] reported an excellent review of the cannabinoid system in regulating normal and inflamed intestines, the colorectal cancer properties of CB1 and CB2 receptor cannabinoids, and their role in colorectal cancer pathogenesis, prevention, and treatment. In primary cell culture, CB1 and CB2 receptors have been reported to be expressed at a higher level in pancreatic β cells compared with non-β cells [7].”

  1. McAllister, S.D.; Abood, M.E.; Califano, J.; Guzmán, M. Cannabinoid Cancer Biology and Prevention. J Natl Cancer Inst Monogr 2021, 2021, 99-106, doi:10.1093/jncimonographs/lgab008.
  2. Donadelli, M.; Dando, I.; Zaniboni, T.; Costanzo, C.; Dalla Pozza, E.; Scupoli, M.T.; Scarpa, A.; Zappavigna, S.; Marra, M.; Abbruzzese, A., et al. Gemcitabine/cannabinoid combination triggers autophagy in pancreatic cancer cells through a ROS-mediated mechanism. Cell Death Dis 2011, 2, e152, doi:10.1038/cddis.2011.36.
  3. Cherkasova, V.; Kovalchuk, O.; Kovalchuk, I. Cannabinoids and Endocannabinoid System Changes in Intestinal Inflammation and Colorectal Cancer. Cancers (Basel) 2021, 13, doi:10.3390/cancers13174353.
  4. Bunsick, D.A.; Matsukubo, J.; Aldbai, R.; Baghaie, L.; Szewczuk, M.R. Functional Selectivity of Cannabinoid Type 1 G Protein-Coupled Receptor Agonists in Transactivating Glycosylated Receptors on Cancer Cells to Induce Epithelial-Mesenchymal Transition Metastatic Phenotype. Cells 2024, 13, doi:10.3390/cells13060480.
  5. Barajas-Martínez, A.; Bermeo, K.; de la Cruz, L.; Martínez-Vargas, M.; Martínez-Tapia, R.J.; García, D.E.; Navarro, L. Cannabinoid receptors are differentially regulated in the pancreatic islets during the early development of metabolic syndrome. Islets 2020, 12, 134-144, doi:10.1080/19382014.2020.1849927.

  1. Also in the introduction, paragraph 2 discusses processes that synthetic cannabinoids have been shown to affect. These seem to support that there are various other receptors that may be involved, in which the authors do not elaborate.

Author response: Thank you for the comment. We have added the following in the introduction: “ Interestingly, in addition to CB1 and CB2, there are several other receptors, such as GPR119, GPR55 [8], peroxisome proliferating activated receptor a (PPARa), and PPARg [9], that cannabinoids may respond to [10].”

  1. Ryberg, E.; Larsson, N.; Sjögren, S.; Hjorth, S.; Hermansson, N.O.; Leonova, J.; Elebring, T.; Nilsson, K.; Drmota, T.; Greasley, P.J. The orphan receptor GPR55 is a novel cannabinoid receptor. Br J Pharmacol 2007, 152, 1092-1101, doi:10.1038/sj.bjp.0707460.
  2. Sun, Y.; Bennett, A. Cannabinoids: A New Group of Agonists of PPARs. PPAR Research 2007, 2007, 023513, doi:https://doi.org/10.1155/2007/23513.
  3. Zhang, M.W.; Ho, RC The Cannabis Dilemma: A Review of Its Associated Risks and Clinical Efficacy. J Addict 2015, 2015, 707596, doi:10.1155/2015/707596.

  1. The sentence “CB1, a primary target for synthetic cannabinoids, can cause diverse cellular responses depending on the agonist used [8].” Is vague – what is meant by this? How many different agonists are there for CB1 receptors, and how variable are these responses? It would also be nice to discuss the rationale for using the neuraminidase inhibitors chosen – is there any data that these affect cancer cells and/or signaling from CB1 receptors?

Author response: Thank you for the comment. We have removed the sentence and added new concepts in the introduction. We have reported on the rationale of using neuraminidase inhibitors such as BIM23, MMP9i and oseltamivir phosphate (OP), which function on the signaling platform as previously reported by us:

Bunsick, D.A.; Matsukubo, J.; Aldbai, R.; Baghaie, L.; Szewczuk, M.R. Functional Selectivity of Cannabinoid Type 1 G Protein-Coupled Receptor Agonists in Transactivating Glycosylated Receptors on Cancer Cells to Induce Epithelial-Mesenchymal Transition Metastatic Phenotype. Cells 2024, 13, doi:10.3390/cells13060480.

David A. Bunsick , Jenna Matsukubo and Myron R. Szewczuk * Cannabinoids Transmogrify Cancer Metabolic Phenotype via Epigenetic Reprogramming and a Novel CBD Biased G Protein‐Coupled Receptor Signaling Platform. Cancers 2023, 15, 1030. https://doi.org/10.3390/cancers15041030

  1. In the introduction, the authors jump from discussing NF-κB and API (AP-1?) signaling involving Neu-1 and MMP-9, to then discussing DNA methylation, and then back to NF-κB pathways, which made for a confusing introduction. After reading the introduction, I lost sight of the main point of the manuscript was to show data on cancer cell migration/EMT. The authors do not show any data on signaling pathways or epigenetic regulation, yet the introduction leads the reader this direction Their last paragraph clearly states “Here, we investigated the effects of synthetic CB1 cannabinoids on EMT markers, cell viability, migration and tunneling nanotubes signaling pathways in PANC-1 and SW-620 cancer cell lines.”

Author response: Thank you for the comment. We have revised the introduction accordingly.

  1. All figure legends are written like a rehash of the methods sections and need to be edited.

Author response: Thank you for the comment. We have published many peer-reviewed research articles using the descriptives in the figure legends. It is important to describe the actual details of the data in the figure legend. Please see the following peer-reviewed research we published as a follow-up to this study:

Bunsick, D.A.; Matsukubo, J.; Aldbai, R.; Baghaie, L.; Szewczuk, M.R. Functional Selectivity of Cannabinoid Type 1 G Protein-Coupled Receptor Agonists in Transactivating Glycosylated Receptors on Cancer Cells to Induce Epithelial-Mesenchymal Transition Metastatic Phenotype. Cells 2024, 13, doi:10.3390/cells13060480.

 .

  1. Figure 1: Figures 1A-B are at the bottom, and there is no mention of Figures 1D-F in the legend. The TNT images are hard to decipher; what, specifically is being measured? The nanotubes between cells are not evident (to me), nor is there any mention in the introduction that cancer cells produce TNTs, the significance of them, and why these cells would produce them in response to cannabinoids. Figure 1A and 1C CellMask + DAPI images look identical, except enhanced for brightness. Are these images altered?

Author response: Thank you for the comment. We have corrected the legend accordingly (D-F). The images were enhanced (TNT images), and the TNT image cell projections were differentiated on ImageJ. The projections were quantified using Corel Photo-Paint. (C) Representative Images and (D-F) quantification of TNTs of PANC-1 cells treatment with 300 µg/mL oseltamivir phosphate (OP) for 15 minutes before synthetic cannabinoid treatment.”  In the introduction, we have described the rationale of the tunneling nanotube assay used in the study: TNTs are intercellular conduits among cancer, normal, and immune cells to facilitate the transfer of calcium waves, mitochondria, lysosomes, and proteins, which in turn contribute to the survival, metastasis, and chemoresistance in cancer cells [26].”

  1. Melwani, P.K.; Pandey, B.N. Tunneling nanotubes: The intercellular conduits contributing to cancer pathogenesis and its therapy. Biochimica et Biophysica Acta (BBA) - Reviews on Cancer 2023, 1878, 189028, doi:https://doi.org/10.1016/j.bbcan.2023.189028.

Figure 1A and 1C CellMask + DAPI images look identical, except for enhanced brightness. Are these images altered? No alterations! The experiment with the OP treatment was done together and at the same time with the CB1 agonists treatment. 

The figures have been corrected so that they are in proper order. The cell size, shape, and projections are being quantified through the cell mask. The cellular projections are present on the second column of images (Enhanced image). Small needle-like projections can be seen extending from cell to cell and connecting them. Larger projections can also be seen connecting each cell. Since this experiment is only quantifying the presence of these projections and not their identity, we have removed the mention of TNT until the discussion as further analysis is needed on the projections to confirm their identity. The images were only modified to reduce background ‘noise’ from the microscope. No additional changes were made to the images or fluorescence of the CellMask.

  1. Did the authors perform the same experiments as shown in Figure 1 for SW-620 cells? Why were these data not included? If they were done and not significant, then that should be stated at the very least. If they were done previously, the results should be stated here and cited.

Author response: Thank you for the comment. The same experiments were conducted on the SW-620 cell line. However, the size of the cell and zoom of the microscope were too small to determine any changes to the morphology and any cellular projections. As a result, no images were included. However, a section has been added to the discussion mentioning this.

  1. Figure 1C: the IF image of the cells after OP treatment using CellMask are not very uniform. Are the cells still viable, or did the OP treatment result in a substantial number of dead cells? They do not look very healthy, although hard to say if they are dead or have just lifted from the dish. Data should be provided to determine health of the cells to better interpret the data.

Author response: Thank you for the comment. These results are interesting and reportable. We have shown data on the metabolic cell viability using CB1 agonist followed by OP, MMP9i and BIM23 inhibitors. “The addition of synthetic CB1 cannabinoids significantly increased cell viability compared with the untreated control for PANC-1 cells (Figure 2 A-C) and SW-620 cells (Figure 3 A-C). When they were combined with the inhibitors, the cell viability decreased significantly with all three synthetic cannabinoids and the positive control (media containing 10% FBS) (Figure 2 D-F, and H) and (Figure 3 D-F, and H). Cell viability of the PANC-1 cells was not affected by the CB1 cannabinoids with the untreated controls (Figure 2G), while the SW-620 cells were slightly affected. Additionally, cell viability for both cells remained at 80-95% with the inhibitors alone, similar to the untreated control, indicating that using these inhibitors did not result in non-specific adverse cell death (Figure 2G) but slightly for SW-620 cells (Figure 3G). “

Interesting to note is that OP treatment with CB1 agonists reduced TNT projections on remaining viable cells:

We used the data from the AlamarBlue cell viability assay to determine the health of the cells following OP treatment. The results of the experiment showed that the viability of the cell was similar to that of the untreated control. Moreover, it suggests that the presence of OP does not reduce the cell viability any more than cells present in DMEM alone (no growth factors present). It is also possible that some of these cells may have lifted off the dish.

  1. Figure 1 regarding TNTs: Section 2.6 in the methods does not describe the TNT assay other than to mention that CellMask was used to delineate the cell membrane, so the TNT images appear to be outlines of the plasma membrane. Is the total fluorescence of the TNT images being quantitated? This may be over interpreting the effects of treatments as OP treatment reduced cell size, so would naturally reduce the fluorescence of the cell membrane. This is either not clear or misleading. What projections are being measured, exactly? And what does each data point in Figures 1B, D-F represent (what is the sample size/experimental number)?

Author response: Thank you for the comment. We have answered this in the previous question. Based on the experiment, OP reduces cell size and therefore projections and viability. We completed 3-4 experiments where each data point was replicated.

  1. Figures 2 and 3: The y-axes are “Cell Viability Percentage” yet most of the values are over 100%. What does this mean? How does cell viability go from 100% to 250% between untreated cells and cells with FBS (as shown in Figure 3A, for an example). Is there any difference between cells with FBS vs AM-404 in Figure 3A (and same question for other treatments)? There is no mention of why there is not a dose-dependent response (i.e., not consistent changes in viability depending on dose) for some of the figures (e.g., Figures 2B-C and Figures 3A and 3C). Were all comparisons done just between the control and each treatment? Also for these figures, in the methods it is stated that the experiments were done in triplicate, but the legends read “3 independent experiments were performed in triplicate” and the figures have 9 separate data points per treatment. Did you not average the technical replicates or are you treating those as independent data points? This may skew your results.

Author response: Thank you for the comment. An AlamarBlue percentage over 100% typically indicates that the cell population being measured exhibits greater metabolic activity or cell viability compared to the untreated control. The AlamarBlue assay measures cellular metabolic activity based on the reduction of resazurin to resorufin, which produces a fluorescent signal. A value of 100% is representative of the untreated control, whereas values above 100% indicate increased activity or cell proliferation. Cell viability likely increases between the untreated control and FBS group as the untreated control is DMEM (without growth factors), whereas media containing FBS has growth factors present. As a result, the cells with the growth factors present are more likely to be viable compared to cells without any present. A mention of the lack of dose response relationship has been added to the discussion. The figures have been changed to include the average of technical replicates.

  1. Figure 4: I don’t see much difference in expression of E- or N-cadherins in the IF images. Can better images that represent the data be provided? Did the authors look at vimentin expression (or any other EMT markers) that may show more convincing results?

Author response: Thank you for the comment. After extensive peer review, we have reported on the expression of EMT markers using SW-620 colorectal cancer cells (Bunsick et al. Cells 2024, 13, 480.https://doi.org/10.3390/cells13060480): “Using SW-620 cells, there was no statistically significant reduction in E-cadherin on SW-620 cells following CB1 agonist treatment when compared to the control group. However, there is a slight reduction in E-cadherin expression following the treatment. There are two potential reasons for these results. Firstly, the CB1 agonists may have no significant impact on E-cadherin levels. Secondly, given that SW-620 cells have already undergone the EMT process for invasiveness, there may have already been a significant reduction in E-cadherin, and further treatment with the agonists has no further effect on the marker. Despite this, treatment with the CB1 agonists did not increase E-cadherin expression, indicating that these synthetic cannabinoids may not have the tumor suppression function that was once thought. While the majority of research indicates that a decline in E-cadherin expression is a characteristic feature of EMT, the recent literature contends that the absence of E-cadherin is neither causal nor necessary for EMT [63,64]. As such, the study findings are supported by the literature, providing further insight into the specific effects of CB1 agonists on a characteristic feature of EMT marker expression.”

  1. Hollestelle, A.; Peeters, J.K.; Smid, M.; Timmermans, M.; Verhoog, L.C.;Westenend, P.J.; Heine, A.A.; Chan, A.; Sieuwerts, A.M.; Wiemer, E.A.; et al. Loss of E-cadherin is not a necessity for epithelial to mesenchymal transition in human breast cancer. Breast Cancer Res. Treat. 2013, 138, 47–57. [CrossRef]
  2. Chen, A.; Beetham, H.; Black, M.A.; Priya, R.; Telford, B.J.; Guest, J.; Wiggins, G.A.; Godwin, T.D.; Yap, A.S.; Guilford, P.J. E-cadherin loss alters cytoskeletal organization and adhesion in non-malignant breast cells but is insufficient to induce an epithelial-mesenchymal transition. BMC Cancer 2014, 14, 552. [CrossRef]

We have articulated this issue in the results section as follows: “The SW620 cell line was derived from a colon cancer lymph node metastasis (Duke’s type C) and has a rounded or spindle-shaped fibroblast-like morphology [48]. To investigate the effect of the synthetic CB1 cannabinoids on E- and N-cadherin expressions in pancreatic PANC-1 cancer cells, we first investigated the presence of E-cadherin following treatment, similarly to the SW-620 cells. PANC-1 cancer cells are derived from the primary metastatic tumor from a patient (Lieber M, Mazzetta J, Nelson-Rees W, et al. Establishment of a continuous tumor-cell line (panc-1) from a human carcinoma of the exocrine pancreas. Int J Cancer. 1975;15:741–747. [PubMed] [Google Scholar]). PANC-1 was cultured from a patient with an adenocarcinoma in the head of the pancreas, which invaded the duodenal wall. Metastases in one peripancreatic lymph node are deemed evidence during a pancreaticoduodenectomy. Following a 24-hour treatment with the synthetic AM-404, Arvanil, and Olvanil cannabinoids, the data depicted in Figures 4 A and B show that each of the CB1 cannabinoids significantly reduced E-cadherin expression compared to the untreated control. Here, further investigation is required to elucidate the characteristic properties of metastatic cancer cells influenced by the activation of cannabinoids via their functional selectivity [6].

We have enlarged the images in the Figure to show the staining of E- and N-cadherin markers.

  1. Figure 5: While the data show an increase in SEAP expression in macrophages (and why this cell line was used, other than for this particular protocol?), this could be due to upregulation of NF-κB or AP-1, among others, potentially. A confirmatory assay to show evidence of specific NF-κB upregulation is needed.

Author response: Thank you for the comment. The immunofluorescence technique, as recommended to confirm, is quite challenging and can present significant bias, such as reaction bias and interruption bias [Biomark Insights 2010, 5, 9–20]. Reaction bias is described as pitfalls at the technical level, such as specimen fixation, tissue processing, detection system, and tissue pretreatment. Similarly, pitfalls in the design and interpretation level are interpretation bias, which includes the specificity and sensitivity of the antibodies. In addition, this technique has the possibility of having a high background and species cross-reactivity. Also, there are isotype subunits of NFkB: NF-κB subunits p50, p65, and p75.

            The NF-kB complex is sequestered in a dimer form (with p65/p50 dimers as the most common composition) in the cytoplasm, where IkB binds it. Upon activation by various stimuli, IKK interacts with the inhibitory IkB, resulting in phosphorylation, ubiquitination, and degradation of IkB, rendering the dimer free to translocate to the nucleus. Here, the dimer binds to kappa B binding sites of several gene targets that may be involved in cell survival in cells or inflammatory pathways. Dimers consisting of p65/p50 tend to be transcriptionally active, whereas p50 homodimers tend to suppress gene transcription.

          NFkB signaling by NFkB-dependent secretory alkaline phosphatase (SEAP) activity involves the complex signaling pathways as described above in addition to the activation of AP-1. Upon stimulation, RAW-Blue™ cells

activate NF-κB and AP-1, leading to SEAP secretion, detectable and measurable using QUANTI-Blue™ (InvivoGen) SEAP in the medium. RAW-Blue™ cells are made to be resistant to Zeocin™ and G418 antibiotics.

          We have previously published an extensive peer-reviewed article on CB1 agonists using the     NFkB signaling by NFkB-dependent secretory alkaline phosphatase (SEAP) without doing NFkB nuclear localization by immunofluorescence (Bunsick et al. Cells 2024, 13, 480. https://doi.org/10.3390/cells13060480).

  1. Figure 6: the images of the scratch assays are hard to see, and it appears that some of the images are at higher magnification than others. How was the rate of migration measured in these images? Normally it is based on rate of wound closure, yet in some of these images, some cells migrated quite rapidly and others didn’t, so there is not a clear “margin” on either side to measure width of the wound. For example, in Figure 6B control at 12 hours, the wound looks more closed in the image than what is shown at 12 hours for Olvanil, yet the graph indicates that Olvanil has a higher wound closure rate. Having an outline of what was measured in the images may help, but I’d suggest re-evaluating your data to ensure the data are being reported accurately and the conclusions you make in the discussion support the data.

Author response: Thank you for the comment. Imaging was taken with a Nikon Eclipse Ti2 microscope (4x magnification) every hour for the first 6 hours and at 12 after the creation of the scratch wound. We used the 4x magnification to capture the majority of the scratch wound. The wound width was precisely measured at 6-8 points per image using the microscope NIS-Elements AR software and analyzed to create a simple linear regression (red dashed line) using GraphPad Prism 10 to measure the rate of wound closure represented by μm/hr. The quantified data represent two to three independent experiments displaying similar results.

The simple linear regression fits a straight line through the data to find the best-fit value of the slope and intercept. The slope of the linear regression line is the rate of wound gap closure in μm/hr ± standard error of the slope.

(a) Normally it is based on rate of wound closure, yet in some of these images, some cells migrated quite rapidly and others didn’t, so there is not a clear “margin” on either side to measure width of the wound. For example, in Figure 6B control at 12 hours, the wound looks more closed in the image than what is shown at 12 hours for Olvanil, yet the graph indicates that Olvanil has a higher wound closure rate.

Here, we take 6-8 measured points across the wound gap to get the best representative measurement of the wound gap. Each point on the graph is the mean± error of the mean for each time point. The important part of the measurement is that we take the exact picture of the wound gap as illustrated in the Figure. At 12 hours for Olvanil, the wound gap closure rate was higher than the control. These data are reproducible in 3 independent experiments.

  1. In the discussion, paragraph 4, the authors mention that these synthetic cannabinoids reduce E-cadherin and increase N-cadherin and vimentin. No data were shown regarding vimentin, and there are many reports that indicate E- and N-cadherin expression levels are not the only (or best) markers to demonstrate an EMT process. Additionally, drawing conclusions between PANC-1 cells where data are provided in this manuscript to SW-620 cells (which the data may have been reported in reference #3) and draw conclusions about the impact of Olvanil on EMT between these two cells does not allow the reader to determine if this conclusion is accurate. The data provided in Figure 4 are not very convincing; I’d recommend providing protein expression data and/or additional EMT markers to support their conclusions.

Author response: Thank you for the comment. We corrected and removed vimentin in the discussion section, as mentioned. In the discussion, it is appropriate to discuss the findings previously reported using SW-620 cells. This is a standard scholarly discussion, which is necessary and appropriate.

After extensive peer review, we have published before using the same experimental approach ((Bunsick et al. Cells 2024, 13, 480. https://doi.org/10.3390/cells13060480). The data are reproducible in three independent experiments

Additional, minor things to address:

  1. The use of the word “surreptitiously” is misleading (title and in the manuscript) – are the cancer cells being “secretive, avoiding notice or attention”? 
  2. The title is a mouthful and hard to interpret – I’d suggest a rewrite to be more concise/clear.

Author response: Thank you for the comment. Other research articles use “surreptitious” in describing cancer behavior.

  1. a) P Kuppusamy · 2013 · Cited by 23 — “The breaking surreptitious thing of the cancer-related node is still not yet to be found.” Journal of Pharmacy Research

Volume 6, Issue 8, August 2013, Pages 884-892.

  1. b) May 23, 2017 — “The findings, published May 22 in Nature Neuroscience, underscore the surreptitious nature of cancer, which uses a variety of tricks to evade.. “
  2. c) Jun 26, 2013 — “One of the dangerous features of this cancer is the surreptitious nature of early disease.” https://doi.org/10.1111/cas.12201.

The title illustrates precisely the findings of the study. The other three peer-reviewers of the manuscript had no issues with the title. Indeed, by allowing insured secretive communication between cancer cells themselves and with the neighboring stromal cells, tunneling nanotubes (TNTs) are involved in the multistep process of cancer development from tumorigenesis.  

Comments on the Quality of English Language. Please edit the manuscript for grammatical errors and consistency in writing throughout.  

Author response: Thank you for the comment. We use Grammarly software to check for the quality of the written style. It checks for correctness, clarity, engagement and delivery of written style. It also checks for plagiarism.

Submission Date

18 September 2024

Date of this review

01 Oct 2024 23:10:12

Round 2

Reviewer 1 Report

Comments and Suggestions for Authors

Dear authors,
As the manuscript in its revised form also shows strong interpretative and factual weaknesses with regard to the conclusions drawn from the results, I regret that I cannot recommend it for publication.

Author Response

The manuscript, in its revised form, also shows strong interpretative and factual weaknesses regarding the conclusions drawn from the results, so I regret that I cannot recommend it for publication.

Authors' response: We are sorry that this reviewer is not supportive. The other three peer reviewers were very supportive and recommended acceptance after minor revisions.

Reviewer 2 Report

Comments and Suggestions for Authors

Although some parts seems to be insufficient, this improved manuscript is acceptable for publication.

Author Response

Although some parts seem to be insufficient, this improved manuscript is acceptable for publication.

Thank you for your support.

Prof. Dr. Szewczuk

Reviewer 3 Report

Comments and Suggestions for Authors

I checked the revised manuscript. Unfortunately, most of the issues were not addressed.

Author Response

I checked the revised manuscript. Unfortunately, most of the issues were not addressed.

Author response: Thank you for your comment. We will revise the manuscript as suggested.

In this study the authors aimed at investigating the effect of CB1 cannabinoids, specifically AM-404, Arvanil, and Olvanil, on PANC-1 and SW-620 cells in terms of modulating tunneling nanotubes, cell viability and migration, and on some EMT markers.

The manuscript is clearly written and the results clearly presented. However, some important issues influence the conclusions that, in my opinion, are not fully supported by the presented results.

Materials and Methods, paragraph 2.9 and relative results: for the scratch test the authors report that “The control cells were then supplemented with a medium containing 5% FBS. At the same time, the CB1 agonist-treated cells were supplemented with AM-404, Arvanil, or Olvanil in a medium without FBS”. The test should be performed using the dame FBS % (serum free) to have the same cell proliferation and to avoid the influence of cell proliferation on the result. If I understood correctly, control samples were cultured in presence of FBS. If so, the results on the migration are inconsistent.

Results, paragraph 3.3 and figure 4: the authors report an effect of the treatment on the expression of E-cadherin and N-cadherin. The micrographs in figure 4 are too small and some insets or higher magnification micrographs should be included. Moreover, an important point is that the text and the figure legend do not describe the pattern of expression and the localization of the two cadherin. Especially for E-cadherin, is fundamental the description of its localization, that is if the protein is expressed at the cell plasma membrane (cell junction functional) or in the cytoplasm (cell junction degraded and not functional). The quantitative analysis is not sufficiently informative.

We have added the following  in the results section to explain this issue: "It is noteworthy that  CB1 agonist-induced internalization of G-protein-coupled receptors is a regulatory mechanism for CB1 receptor abundance and availability at the plasma membrane [53,54]. CB1 receptor trafficking is dynamically regulated by cannabimimetic drugs whereby recovery of CB1 to the cell surface after 20 min but not 90 min after agonist treatment was  reported to be independent of new CB1 receptor synthesis [54]. As depicted in Figure 4, the pattern of expression and the localization of the E- and N-cadherins following CB1 agonists treatment for 24 hr is elucidated by fixing and permeabilizing the cells to detect the proteins  expressed at the cell plasma membrane (cell junction functional) and in the cytoplasm (cell junction degraded and not functional). Glogauer and Blay [55] reviewed the dynamic diversity in cancer cell’s responses to CB1 and CB2 cannabinoids in their invasive and metastatic capacities."

Moreover, figure 4 shows the immunofluorescence analysis of E-and N-cadherin expressions in response to CB1 cannabinoids in PANC-1 and the same results obtained on SW620 cells were published by the same authors (Cells. 2024 Mar; 13(6): 480). Please comment this and clearly highlight the novelty of the results compared to the already published paper.

We have added the following in the discussion to highlight the novelty of these results: "To further substantiate our findings in our previous study [6], we hypothesized that CB1 agonists would affect the expression of EMT E-cadherin and N-cadherin markers in PANC-1 cells. We found that all three synthetic CB1 cannabinoids significantly reduced E-cadherin expression and increased N-cadherin (Figure 4). These results suggest that Olvanil has a more significant impact on metastasis in PANC-1 cells than on SW-620 cells, as reported by Bunsick et al. [6]. For example, although all three CB1 agonists upregulated vimentin expression in SW-620 cells, Olvanil had a weaker and non-significant effect on vimentin expression. The novelty of these findings signifies that CB1 agonist strength may impact the expression of vimentin, suggesting a kinetic component within the CB1 receptor that needs to be studied further. To explain these results, there are different cannabinoid allosteric ligands with different degrees of modulation, called ‘biased modulation,’ that can vary dramatically in a probe- and pathway-specific manner. The response seen here may be a result of specific CB1 functional selectivity, not from differences in orthosteric ligand efficacy or stimulus-response coupling. Another interesting result is that Arvanil had a more substantial effect on vimentin expression (SW-620) [6] than on N-cadherin expression (PANC-1), which may indicate that Arvanil has a weaker impact on PANC-1 cells than on SW-620 cells. "

Reviewer 4 Report

Comments and Suggestions for Authors

1. Thank you for editing the introduction to include the distribution of CB1 receptors outside of the CNS. This enhances the significance of the study, in my opinion. The other edits to the introduction are good and appreciated.

2. Figure legends should provide detail, and the Cells (2024) paper you referenced does have an adequate description of the figures, as do other articles in Cells by other authors.  However, if you compare the level of detail in Figure 1 legend in this manuscript vs your Figure 3 legend in the paper you reference (doi:10.3390/cells13060480), it is above the top in this manuscript. You did not go into such detail as using "a 24-well plate on 12 mm glass coverslips", etc in your previous paper. You can trim down the details that should be within the methods section and retain the necessary details in the legends to allow the reader to interpret your results, as you did in the 2024 paper.  However, I leave this up to the discretion of the editorial staff.

3. The authors addressed my comment #7 with this: The same experiments were conducted on the SW-620 cell line. However, the size of the cell and zoom of the microscope were too small to determine any changes to the morphology and any cellular projections. As a result, no images were included. However, a section has been added to the discussion mentioning this. Yet, I don't see that the discussion was edited as mentioned (i.e., no highlighted additional text is provided). 

4. Thank you for the clarification on OP treatment (for my comment #8). This was my mistake in interpreting your data in Figure 2, but you did see a significant reduction in SW-620 cell viability with OP treatment compared to untreated controls (Figure 3G). Saying "We used the data from the AlamarBlue cell viability assay to determine the health of the cells following OP treatment. The results of the experiment showed that the viability of the cell was similar to that of the untreated control." is true for the PANC-1 cell line but not the SW-620 cell line; I wouldn't say this was "similar" but is a statistically significant result. 

5. I still do not fully understand the TNT measurements and number of replicates to address my comment #9. Your response of "Based on the experiment, OP reduces cell size and therefore projections and viability. We completed 3-4 experiments where each data point was replicated." is unclear and contradicts your earlier comment where OP treatment does not affect viability differently than untreated controls. 

6. Thank you for clarifying the AlamarBlue viability assay.

7. Your E-cadherin and N-cadherin images are much better and easier to visualize and interpret; thank you for updating these. I would have liked to see other markers shown, such as vimentin, as it is known that E-cadherin expression levels do not always correlate well to EMT, which would only serve to strengthen the author's data. 

8. To address my comment #12, the authors thoroughly discuss the biases of this assay, and then conclude that "We have previously published an extensive peer-reviewed article on CB1 agonists using the     NFkB signaling by NFkB-dependent secretory alkaline phosphatase (SEAP) without doing NFkB nuclear localization by immunofluorescence."  I leave this to the editorial staff to determine if the additional confirmatory experiment should be performed to address my concern.

9. Thank you for addressing my comment #13 regarding the scratch assays and how you measured the wound closure. Randomly measuring 6-8 points may have introduced some needless bias as some cells may have remained in the gap following scratching and, if included in the measurements, this should be considered extraneous. There was no mention of this, so I'm assuming you did this accurately. I still have concerns, but I will rely on other reviewers and the editors to determine if the explanation is adequate for publication in Cells. 

10. I agree it is appropriate to discuss previously reported findings, and I did not intend to imply otherwise. My comment was meant for the authors to either include the data from a previous paper (especially since it is your data!) or summarize the findings, instead of just including the citation and having the reader go find the information; reading this paper alone should be sufficient to allow the reader to determine if conclusions being made regarding a specific cell type discussed were appropriate. 

11.  Yes, other papers correctly use the term "surreptitious" - especially the ones you cited that are specifically referring to how "tricky" cancer cells can be.  Your data, unless I am mistaken on the point of your paper, is illustrating how cannabinoids can induce cellular communication by TNTs and inducing an EMT process. A recent article (Valdebenito, et al. 2021, doi.org/10.1038/s41598-021-93775-8 states "...TNTs are an efficient cell-to-cell communication system used by cancer cells to adapt the microenvironment to the invasive nature of the tumor." My point was that TNTs and EMT are well known processes by cancer cells (and other cells). 

12. "The title illustrates precisely the findings of the study. The other three peer-reviewers of the manuscript had no issues with the title."  My comment about the title was that it was unwieldly, not that it is incorrect. I read long, cumbersome titles and that turns me off of even reading the article.  I accept that I may be alone in that opinion, but if you can shorten it and be clear, I believe that would improve your readership. Something as simple as: Synthetic CB1 Cannabinoids Promote Tunneling Nanotube Communication, Cellular Migration, and EMT in Pancreatic PANC-1 and Colorectal SW-620 Cancer Cell Lines".  Just a thought.

Comments on the Quality of English Language

Please do not rely on Grammarly software as your only source of editing. While it is a good software program, it is not perfect and still makes some mistakes. 

Author Response

  1. Thank you for editing the introduction to include the distribution of CB1 receptors outside of the CNS. This enhances the significance of the study, in my opinion. The other edits to the introduction are good and appreciated.
  2. Figure legends should provide detail, and the Cells (2024) paper you referenced does have an adequate description of the figures, as do other articles in Cells by other authors.  However, if you compare the level of detail in Figure 1 legend in this manuscript vs your Figure 3 legend in the paper you reference (doi:10.3390/cells13060480), it is above the top in this manuscript. You did not go into such detail as using "a 24-well plate on 12 mm glass coverslips", etc in your previous paper. You can trim down the details that should be within the methods section and retain the necessary details in the legends to allow the reader to interpret your results, as you did in the 2024 paper.  However, I leave this up to the discretion of the editorial staff.

Author response: We have trimmed down the details in the figure legends as recommended.

  1. The authors addressed my comment #7 with this: The same experiments were conducted on the SW-620 cell line. However, the size of the cell and zoom of the microscope were too small to determine any changes to the morphology and any cellular projections. As a result, no images were included. However, a section has been added to the discussion mentioning this. Yet, I don't see that the discussion was edited as mentioned (i.e., no highlighted additional text is provided). 

Author response: We missed adding this comment in the discussion. We have now included it highlighted in blue text.

  1. Thank you for the clarification on OP treatment (for my comment #8). This was my mistake in interpreting your data in Figure 2, but you did see a significant reduction in SW-620 cell viability with OP treatment compared to untreated controls (Figure 3G). Saying "We used the data from the AlamarBlue cell viability assay to determine the health of the cells following OP treatment. The results of the experiment showed that the viability of the cell was similar to that of the untreated control." is true for the PANC-1 cell line but not the SW-620 cell line; I wouldn't say this was "similar" but is a statistically significant result. 
  2. I still do not fully understand the TNT measurements and number of replicates to address my comment #9. Your response of "Based on the experiment, OP reduces cell size and therefore projections and viability. We completed 3-4 experiments where each data point was replicated." is unclear and contradicts your earlier comment where OP treatment does not affect viability differently than untreated controls. 
  3. Thank you for clarifying the AlamarBlue viability assay.
  4. Your E-cadherin and N-cadherin images are much better and easier to visualize and interpret; thank you for updating these. I would have liked to see other markers shown, such as vimentin, as it is known that E-cadherin expression levels do not always correlate well to EMT, which would only serve to strengthen the author's data. 

Author response: There is evidence that carcinomas with markers of mesenchymal differentiation have different biological and clinical behaviour (Domagala et al, 1990a,1990b; Medeiros et al, 1988; Liu et al, 2010). Pancreatic cancer vimentin expression patterns have been investigated in small series. Schussler et al (1992) reported a lack of expression in pancreatic cancers, while Nakajima et al (2004) observed sparse cell labelling in 30 primary pancreatic adenocarcinomas with prominent expression in 15 liver metastases. Similarly, vimentin labelling was more pronounced in widely metastatic pancreatic cancers as compared with locally destructive ones (Naito and Iacobuzio-Donahue, 2010). The main focus of our study is to see how CB1 agonist affect the EMT markers. For SW-620 cells, they do not express N-cadherin and so we used vimentin. However, for pancreatic cancer cells, we were able to measure N-cadherin to support our hypothesis.

  1. To address my comment #12, the authors thoroughly discuss the biases of this assay, and then conclude that "We have previously published an extensive peer-reviewed article on CB1 agonists using the     NFkB signaling by NFkB-dependent secretory alkaline phosphatase (SEAP) without doing NFkB nuclear localization by immunofluorescence."I leave this to the editorial staff to determine if the additional confirmatory experiment should be performed to address my concern.

Author response: We have previously explained the rationale why we used the SEAP assay. We have published several peer-reviewed articles using this assay system to measure NFkB activity.

  1. Thank you for addressing my comment #13 regarding the scratch assays and how you measured the wound closure. Randomly measuring 6-8 points may have introduced some needless bias as some cells may have remained in the gap following scratching and, if included in the measurements, this should be considered extraneous. There was no mention of this, so I'm assuming you did this accurately. I still have concerns, but I will rely on other reviewers and the editors to determine if the explanation is adequate for publication in Cells.

Author response: Yes, we have done this as accurately as possible. Noteworthy, we have measured several areas of the gap to get a true representation of cell migration, but more importantly, we have done this every hour. We used the inverted microscope software to accurately measure the gap distance when the image was viewed. We have reported scratch wound technique previously (Nutrients 2024, 16, 1840. https://doi.org/10.3390/nu16121840). 

  1. I agree it is appropriate to discuss previously reported findings, and I did not intend to imply otherwise. My comment was meant for the authors to either include the data from a previous paper (especially since it is your data!) or summarize the findings, instead of just including the citation and having the reader go find the information; reading this paper alone should be sufficient to allow the reader to determine if conclusions being made regarding a specific cell type discussed were appropriate. 

Author response: We have highlighted the findings sufficiently. Thank you for the comment.

  1. Yes, other papers correctly use the term "surreptitious" - especially the ones you cited that are specifically referring to how "tricky" cancer cells can be.  Your data, unless I am mistaken on the point of your paper, is illustrating how cannabinoids can induce cellular communication by TNTs and inducing an EMT process. A recent article (Valdebenito, et al. 2021, doi.org/10.1038/s41598-021-93775-8 states "...TNTs are an efficient cell-to-cell communication system used by cancer cells to adapt the microenvironment to the invasive nature of the tumor." My point was that TNTs and EMT are well known processes by cancer cells (and other cells). 

Author response: Yes, the CB1 agonists can induce TNTs and EMT.

  1. "The title illustrates precisely the findings of the study. The other three peer-reviewers of the manuscript had no issues with the title."  My comment about the title was that it was unwieldly, not that it is incorrect. I read long, cumbersome titles and that turns me off of even reading the article.  I accept that I may be alone in that opinion, but if you can shorten it and be clear, I believe that would improve your readership. Something as simple as: Synthetic CB1 Cannabinoids Promote Tunneling Nanotube Communication, Cellular Migration, and EMT in Pancreatic PANC-1 and Colorectal SW-620 Cancer Cell Lines".  Just a thought.

Author response: We have changed the title as suggested. Thank you.

Comments on the Quality of English Language

Please do not rely on Grammarly software as your only source of editing. While it is a good software program, it is not perfect and still makes some mistakes. 

Author response: Yes, Grammarly can make mistakes but we routinely checked this to make sure the text is correct.

Round 3

Reviewer 3 Report

Comments and Suggestions for Authors

I checked the revised manuscript and the new revisions provided by the authors. The manuscript is improved but some issues remain. The authors described how E-cadherin and N-cadherin were investigated, and not if the treatment modified the pattern of expression in terms of localization.

I feel that the authors have not a strong background on EMT since they do not consider that E-cadherin up- or down-regulation and topography as well have different effects in cells having a “more epithelial” or a “more mesenchymal” phenotype. This is a very important point considering that these cells, that have a hybrid phenotype in relation to EMT markers, can invade using two different mechanisms that are single cell invasion or collective invasion. The expression of EMT markers such as E-cadherin should be discussed in relation to this, also considering that PANC-1 are cells having an “epithelial” phenotype. Finally, to compare the effect of cannabinoids on PANC-1 and SW620 the phenotype of cells should be considered.

Author Response

E-cadherin up- or down-regulation and topography as well have different effects in cells having a “more epithelial” or a “more mesenchymal” phenotype. This is a very important point considering that these cells, that have a hybrid phenotype in relation to EMT markers, can invade using two different mechanisms that are single cell invasion or collective invasion. The expression of EMT markers such as E-cadherin should be discussed in relation to this, also considering that PANC-1 are cells having an “epithelial” phenotype. Finally, to compare the effect of cannabinoids on PANC-1 and SW620 the phenotype of cells should be considered.

Author response: Thank you for these excellent comments. We will include these points in the discussion to highlight the EMT cadherin up- or down-regulation and topography as well the effects in cells having a “more epithelial” or a “more mesenchymal” phenotype. This is a very important point, considering that these cells, which have a hybrid phenotype in relation to EMT markers, can invade using two different mechanisms that are single cell invasion or collective invasion. The expression of EMT markers such as E-cadherin will be discussed in relation to this, also considering that PANC-1 are cells having an “epithelial” phenotype. Finally, we will compare the effects of these cannabinoids on PANC-1 and SW620, as well as the phenotype of these cells.

We will also highlight the role played by CB1 agonists in inducing the "cadherin switch," as reported by Cells 2019, 8, 1118; doi: 10.3390/cells8101118  can be added.